# Compulsive alcohol drinking in rodents is associated with altered representations of behavioral control and seeking in dorsal medial prefrontal cortex

Nicholas M. Timme [1✉], Baofeng Ma[1], David Linsenbardt[2], Ethan Cornwell[1], Taylor Galbari[1] & Christopher C. Lapish[1,3]

A key feature of compulsive alcohol drinking is continuing to drink despite negative consequences. To examine the changes in neural activity that underlie this behavior, compulsive alcohol drinking was assessed in a validated rodent model of heritable risk for excessive drinking (alcohol preferring (P) rats). Neural activity was measured in dorsal medial prefrontal cortex (dmPFC—a brain region involved in maladaptive decision-making) and assessed via change point analyses and novel principal component analyses. Neural population representations of specific decision-making variables were measured to determine how they were altered in animals that drink alcohol compulsively. Compulsive animals showed weakened representations of behavioral control signals, but strengthened representations of alcohol seeking-related signals. Finally, chemogenetic-based excitation of dmPFC prevented escalation of compulsive alcohol drinking. Collectively, these data indicate that compulsive alcohol drinking in rats is associated with alterations in dmPFC neural activity that underlie diminished behavioral control and enhanced seeking.

[1] Psychology Department, Indiana University—Purdue University Indianapolis, Indianapolis, IN 46237, USA. [2] Department of Neurosciences, University of New Mexico, Albuquerque, NM 87131, USA. [3] Stark Neurosciences Research Institute, Indiana University—Purdue University Indianapolis, Indianapolis, IN 46237, USA. ✉email: nicholas.m.timme@gmail.com

Continued use of alcohol despite negative consequences (referred to as compulsive[1–3]) characterizes an advanced stage of addiction associated with poor treatment outcomes. Developing novel ways to treat addiction at this stage requires the identification of the brain mechanisms that underlie the decision to compulsively use alcohol. Isolating these mechanisms is difficult as decision-making deficits can exist both as a risk factor for and consequence of addiction[4]. Towards this goal, the current study assessed the decision to drink alcohol in a rodent model of compulsive drinking.

Impairments in decision-making are observed following excessive alcohol exposure[4] and are accompanied by clear changes in structure and function of neurons in medial prefrontal cortex (mPFC)[5–7]. Furthermore, mPFC plays a critical role in many forms of decision-making including drug seeking and extinction[8–10], thus highlighting the importance of this region for expression of pathological drinking. It is not clear how the computational properties of mPFC are altered during addiction, but alcohol (and other appetitive) cues increase neuronal activity in human medial PFC areas[11,12], where activity is linked to perceived reward value[13,14]. In addition, mPFC plays a key role in guiding decision-making during conflict[15,16], and compulsive drinking has been conceptualized as impaired processing of conflict between desire to avoid punishment and drive to consume alcohol[17].

In animal models, compulsive drinking is frequently operationally defined as aversion-resistant drinking (ARD)[1,2]. Rodents with heritable risk for excessive drinking rapidly progress to ARD[18,19], suggesting a pathophysiology in circuits required to control compulsive drinking. In support of this view, previous work in rats bred for excessive drinking, alcohol preferring rats (P rats), identified alterations in mPFC neural activity that reflected a weakened representation of the "intention" to drink alcohol[20], indicating that the role of this brain region in decision-making was compromised in non-compulsive alcohol drinking.

Outbred rats exhibit compulsive drinking following prolonged alcohol exposure[1,2], which is mediated by connections from mPFC to ventral striatum[21]. Also, decreased drive from mPFC to periaqueductal gray contributes to compulsive drinking in mice[22]. In heavy-drinking humans, compulsive responding for future alcohol at the risk of immediate punishment corresponded with increased activity in mPFC during alcohol-associated cues[23]. These mPFC findings synergize with behavioral evidence that compulsive drinking involves a "head down and push" strategy[17] where animals endure the aversive stimulus to obtain alcohol. Therefore, it is critical to determine how altered computational properties of mPFC neural ensembles contribute to expression of compulsive drinking.

To directly examine the importance of mPFC networks for compulsive drinking, we recorded single unit activity from large ensembles of dorsal mPFC (dmPFC, primarily prelimbic and cingulate cortex[24,25]) neurons in rodent models of compulsive drinking and non-compulsive drinking during a simple cued-access protocol task. To assess compulsive (ARD) intake, alcohol was adulterated with the bitter tastant quinine in selected drinking sessions (aversive stimulus, "challenged drinking")[1,2]. Previous work from our group established that following a short (2–4 week) drinking induction, compulsive drinking patterns were consistently observed in alcohol preferring P rats whereas non-compulsive consumption was consistently observed in Wistars[18]. This strong relationship between genotype and phenotype is not surprising given the well-established existence of heritable risk for AUD[26,27] and previous research demonstrating that P rats are a valid pre-clinical rodent model of genetic risk for AUD[28].

P rats were derived from Wistars[29] and Wistars have been shown to drink compulsively with sufficiently long drinking histories[1,2,30–32]. In this study, we used Wistars with short drinking histories to obtain rats that drink alcohol voluntarily, but not compulsively, as a comparison strain to P rats in order to examine the neurocomputational mechanisms of compulsive drinking.

We evaluated neuronal representations of decision-making relevant signals by dmPFC populations during alcohol and alcohol+quinine sessions using principal component analysis (PCA)[33]. To do so, we developed a novel implementation of PCA capable of identifying stable principal components (PCs) across neural populations of varying size. By analyzing populations of neurons at high temporal resolution[34,35] it was possible to examine how large-scale patterns of neural activity evolve during the compulsive decision to drink. We deconstructed the decision-making process into multiple stages by comparing different trials and different time points[36] including pre-trial seeking state, alcohol-availability cues, sipper approach initiation, and drinking. Critically, we also utilized change point analyses[37] to identify changes in animal seeking state and sipper approach likelihood. With these methods, we tested the hypothesis that representations of behavioral control signals are weakened in dmPFC during compulsive drinking.

Finally, emerging data in humans suggests that targeting the mPFC[38–40] could reduce craving and intake in patients diagnosed with a substance use disorder[41,42]. Thus, we used a chemogenetic approach to stimulate neurons in dmPFC to test the hypothesis that increasing dmPFC activity would reduce compulsive drinking.

## Results

**P rats exhibit compulsive drinking**. The 2-Way Conditioned Access Protocol (2CAP) task was previously developed by our group to assess cued drinking with or without aversive consequences[20,43,44]. Following initial alcohol exposure in home cage (Intermittent Access Protocol (IAP), Fig. 1a), P rats and Wistar rats (Wistars) were trained in the 2CAP task (Fig. 1b). Each 2CAP session contained 48 positive conditioned stimulus (CS+) and 48 negative conditioned stimulus (CS−) trials and was conducted in a chamber with two stimulus lights and retractable sippers on opposite ends of the chamber (Fig. 1c, d). On CS+ trials, a single light (blinking or solid, counter balanced across rats) was on for 4 s and indicated the side of the chamber where 10% ethanol fluid would become accessible for 8 s. On CS− trials, both lights were illuminated (opposite modality from CS+) and no access was provided.

Following training in the 2CAP task with 10% ethanol (v/v) only, selected animals were implanted with multielectrode probes (Cambridge Neurotech) in dmPFC. After recovery and task reacclimation, animals underwent a two-day compulsive drinking test sequence that consisted of a baseline regular 2CAP session with 10% ethanol only and a compulsive 2CAP session with 10% ethanol adulterated with 0.1 g/L quinine (Fig. 1a). Electrophysiological data were recorded (OpenEphys) and spike sorted (Kilosort2[45]) for these test sessions. Multiple compulsive drinking tests sequences (baseline and test day) were performed and analyzed. Results for the first compulsive drinking test sequence will be presented first, followed by results for subsequent compulsive drinking tests in later sections. We chose to initially present the first quinine exposure because these sessions were unbiased by possible effects from acclimation to the taste of quinine and to improve clarity in the results by removing compulsive drinking test number as an additional variable. In total, 7 Wistars and 7P rats received electrophysiology probe implants. The initial compulsivity test data for one Wistar was lost, resulting in 6 Wistars and 7P rats for this portion of the analysis (see Methods for full description of number of animals used throughout the experiment).

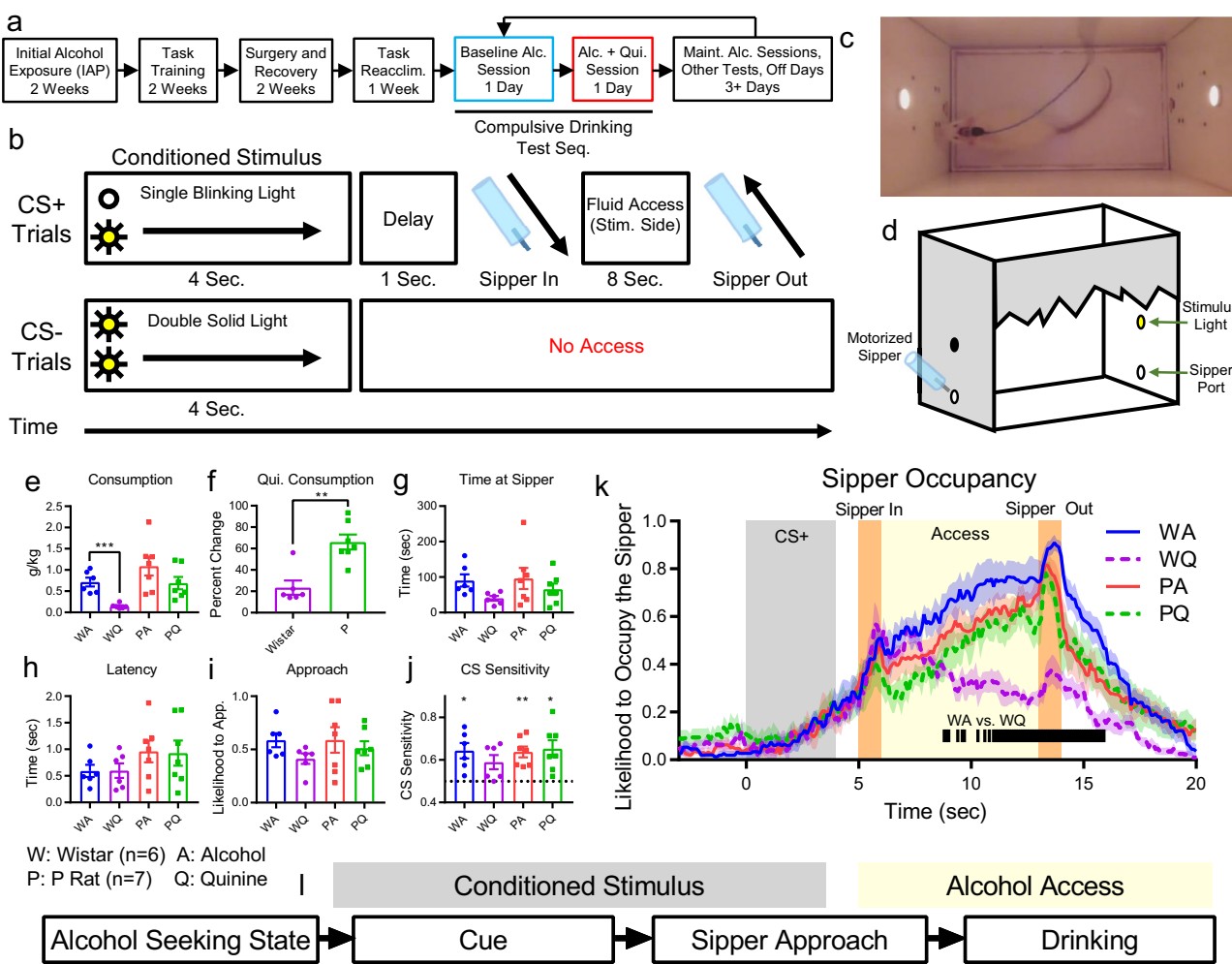

**Fig. 1 P rats exhibit compulsive drinking. a** Experimental schedule for all animals. Electrophysiological data were gathered in two test sessions (compulsive drinking test sequence): 10% alcohol (blue, Tues.) and 10% alcohol + 0.1 g/L quinine (red, Wed.). Compulsive drinking tests were separated by at least 3 days, during which animals underwent alcohol maintenance sessions, other tests (e.g., quinine preference test), or were untested. **b** 2-Way Conditioned Access Protocol (2CAP) task. On CS+ trials, a stimulus light cued the animal to the side of the chamber where fluid would become available. On CS- trials, both stimulus lights were illuminated and modality of the light (blinking vs. solid) was switched. **c** Example image of rat in the chamber with both stimulus lights illuminated. **d** Chamber diagram. **e** Alcohol consumption dropped significantly for Wistars drinking alcohol + quinine (WQ) vs alcohol-only (WA), but not for P rats (PQ vs PA) (***: $p < 10^{-3}$). **f** Percent change in consumption during the alcohol+quinine session relative to the alcohol only session showed a smaller decrease in P rats (**: $p < 10^{-2}$). Time at sipper (**g**), latency to approach the sipper (**h**), and likelihood to approach throughout the session (**i**) did not differ with the addition of quinine. **j** In all but one group, the animals successfully discriminated between CS+ and CS− (CS Sensitivity = CS+ approaches / all approaches, *: $p < 0.05$, **: $p < 10^{-2}$). **k** Likelihood to occupy the sipper (i.e., to be located at the sipper, in the position to drink, attending the sipper) at a given time during approach trials was similar for all groups up to the beginning of fluid access, where Wistars tended to leave the sipper drinking alcohol + quinine (vertical lines denote time bins with FDR-corrected t-test $p < 0.01$ for Wistars drinking alcohol vs. Wistars drinking alcohol + quinine, the corresponding P rat comparisons were not significant). **l** We examined dmPFC representations of four key decision-making variables during different task stages: alcohol seeking state (prior to CS onset), alcohol cue (early CS), sipper approach (later CS), and current drinking (access). Source data are provided as a Source Data file. For all graphs, data are presented as mean +/− sem.

Replicating our previous work[18], we found that Wistars decreased consumption significantly when alcohol was adulterated with quinine (Fig. 1e, strain main effect: F(1,22) = 9.2, $p = 0.0062$, liquid main effect: F(1,22) = 10.05, $p = 0.0044$, post-hoc Bonferroni corrected t-test, Wistar alcohol vs. alcohol+quinine: t(10) = 5.33, $p = 9.9*10^{-4}$). Conversely, P rats showed a smaller reduction in consumption of alcohol-quinine, indicating that P rats exhibited compulsive drinking to a greater degree than Wistars (Fig. 1f, t(11) = 4.39, $p = 0.0011$). P rats exhibited a preference for unadulterated alcohol during a separate alcohol vs. alcohol+quinine choice test (Supplementary Fig. 1) and exposure to quinine at a similar concentration produced elevated aversive facial responses in P rats[46]. These data indicate that P rats could

taste the quinine and found it aversive. Collectively these data support the characterization of P rats as compulsive rats and Wistars as non-compulsive rats and will be referred to as such in subsequent analyses. We wish to emphasize that our primary goal was to compare compulsive and non-compulsive rats based on their behavior. However, these results indicate that this phenotypic split aligned closely with strain, which is not surprising given the well-established role genetic risk plays in AUD[26,27,47].

Though it appeared that Wistars spent less time at the sipper on quinine days (Fig. 1g) and approached the sipper more quickly (Fig. 1h), no significant strain or alcohol-condition differences were found. Also, no differences were observed in the proportion

of trials where the animal approached a sipper (Fig. 1i). All animals were more likely to approach the sipper on CS+ trials relative to CS- trials, indicating that they could differentiate CS+ from CS− (Fig. 1j, Bonferroni corrected one sample t-tests relative to $\mu_0 = 0.5$, Wistar/Alcohol: $t(5) = 3.96$, $p = 0.043$, Wistar/Alcohol+Quinine: $t(5) = 2.68$, $p = 0.18$, P Rat/Alcohol: $t(6) = 5.42$, $p = 0.0064$, P Rat/Alcohol+Quinine: $t(6) = 3.82$, $p = 0.035$). As there was no punishment for approaching on a CS− trial, animals still frequently approached on CS− trials. The reduced consumption and time at sipper in Wistars were driven behaviorally by a tendency in alcohol-quinine sessions to approach the sipper, but then leave soon after arriving (Fig. 1k).

**A method to assess neural population representations**. We examined neural representations in dmPFC of four key decision-making variables by examining epochs of the task (Fig. 1k): a pre-trial period to assess neural representation of the animal's alcohol-seeking state (i.e., seeking or not-seeking), an early CS period to assess neural representation of the alcohol cue, a late CS period to assess neural representation of sipper approach initiation, and an access period to assess neural representation of current drinking. These epochs were defined in relation to task stimuli or using animal behavior data (see below for more details).

We developed a method that used principal component analysis[33] (PCA) to examine the representations[48] of decision-making variables by populations of neurons in dmPFC (Fig. 2a, see Methods for complete analysis work flow). Neural populations are better able to evaluate neural representations than single neurons in brain regions like dmPFC with very heterogeneous individual neuron firing patterns[49]. Throughout the analysis, we compared averaged firing rates across pairs of trial types defined either by environmental stimuli (e.g., CS+ vs. CS−) or behaviorally (e.g., seeking vs. not-seeking, drinking vs. not drinking) to assess neural population representations of a given variable that differed across the trial types. Note that we do not attempt to identify which trial type represents the signal, rather we only quantify separation as a measure of representation strength. In this way, we can compare neural representations between groups to determine if the neural population in one group is more selective for a certain trial type or is more distinct between trial types than in another group. Also, we note that separation between PC trajectories is not necessarily a measure of the subjective strength of a given signal to the animal (i.e., increased PC separation for seeking behaviors does not necessarily imply increased desire for alcohol by the animal). (See Methods for a comparison to neural encoding/decoding analyses.)

We used four example neurons to demonstrate this analysis methodology (Fig. 2b–d). Distances between PC trajectories for two types of trials quantifies how well the population of neurons represents differences between those trials at a given time. When all neurons were ranked by their contribution to a selected PC in the full analysis of all neurons, broad differences between trial types and heterogeneity in response patterns became apparent in the peri-event firing rates of the neurons (Fig. 2e), which motivated the need for a population level analysis to allow for the examination of differences in a few common signals among all the neurons.

When PCA is performed on neural populations of different sizes, there is a risk that the most variance explaining dimensions are mostly driven by the larger population, thus resulting in the spaces from the smaller population being warped to fit the larger one. To address this, we developed a novel method for selecting PCs that relies on subsampling neurons to identify stable PCs across different groups of animals (Fig. 2f, see Methods).

**Alcohol seeking state representation is enhanced in P rats and during challenged drinking**. We assessed the neural representation of the seeking state through differences in firing between seeking and not-seeking states prior to CS+ onset (Fig. 3a). This critical stage of the decision-making process sets the initial conditions of each cued decision and was assessed prior to each trial using sipper approach. The act of approaching represented a necessary behavioral step to obtain alcohol or alcohol+quinine, demonstrating that the animal was seeking to drink. Across animals, the trial-by-trial likelihood to approach gradually decreased throughout a session (Fig. 3b). However, individual animals tended to exhibit a sharp change in approach behavior from a high approach to low approach state (Fig. 3c). We used the largest magnitude change point in the likelihood to approach across trials for each animal as the transition point from seeking to not-seeking (Fig. 3d). No differences were observed between strains or liquid type for session change point (Fig. 3d, Supplementary Fig. 2).

Trial by trial change in average firing rate across all neurons exhibited an increase at the session change point above randomized data (Fig. 3e). After performing PCA on neural firing patterns (see Methods) and sorting neurons based on contribution to PC 1, we found a large shift in which neurons were active at the session change point (Fig. 3f). The specific trajectories of the stable PCs throughout the trials were highly varied through the trial, but most PCs possessed large separations before the CS+ in seeking versus not-seeking trials (Supplementary Fig. 3). We focused on a 2-second-long time window (seeking epoch). The effect size for PC separation was calculated for the seeking epoch and PCs 1, 2, and 4 had the largest separation (Supplementary Fig. 4). The trajectories for these PCs were shifted in population space before and after the session change point, reflecting a representation of the seeking state (Fig. 3g). By comparing PC separations across seeking and not-seeking trials for all stable PCs, we could examine differences in the strength of representations of seeking signals through time (Fig. 3h). We examined the mean PC separation for each 100 ms time bin in the seeking epoch and found that compulsive rats (P rats) possessed stronger baseline seeking representations on alcohol-only days, while challenged drinking (quinine-alcohol) strengthened seeking representations in both strains (Fig. 3i, strain main effect: $F(1,38) = 429$, $p = 2.6*10^{-22}$, liquid main effect: $F(1,38) = 237$, $p = 6.3*10^{-18}$, interaction: $F(1,38) = 27.5$, $p = 6.2*10^{-6}$, post-hoc Tukey's test, Wistar/Alcohol vs. Wistar/Alcohol+Quinine: $p = 3.8*10^{-9}$, Wistar/Alcohol vs. P Rat/Alcohol: $p = 3.8*10^{-9}$, P Rat/Alcohol vs. P Rat/Alcohol+Quinine: $p = 8.7*10^{-8}$). Increased seeking representation strength implies neural firing patterns are more distinct in dmPFC, not necessarily that animals have an increased desire for alcohol or alcohol + quinine. Collectively, these data show increased seeking representation during challenged drinking and in compulsive rats during alcohol drinking.

**Cue representation is diminished during compulsive drinking**. The CS+ alerted animals to trial onset and identified the correct sipper where access would be provided (Fig. 4a). We compared neural firing patterns shortly before the CS+ onset ($-2.4$ to $-0.5$ s) to neural firing patterns immediately following CS+ onset (0 to 1.9 s) for all CS+ trials to examine the representation of CS+ onset. PC trajectories during CS+ showed clear movement through population activity space (Fig. 4b, Supplementary Fig. 5). PC separations rose quickly at CS+ onset and then decayed in all groups (Fig. 4c). When comparing mean PC separation values for each 100 ms time bin in these time windows, challenged drinking increased CS+ onset representation strength in non-compulsive

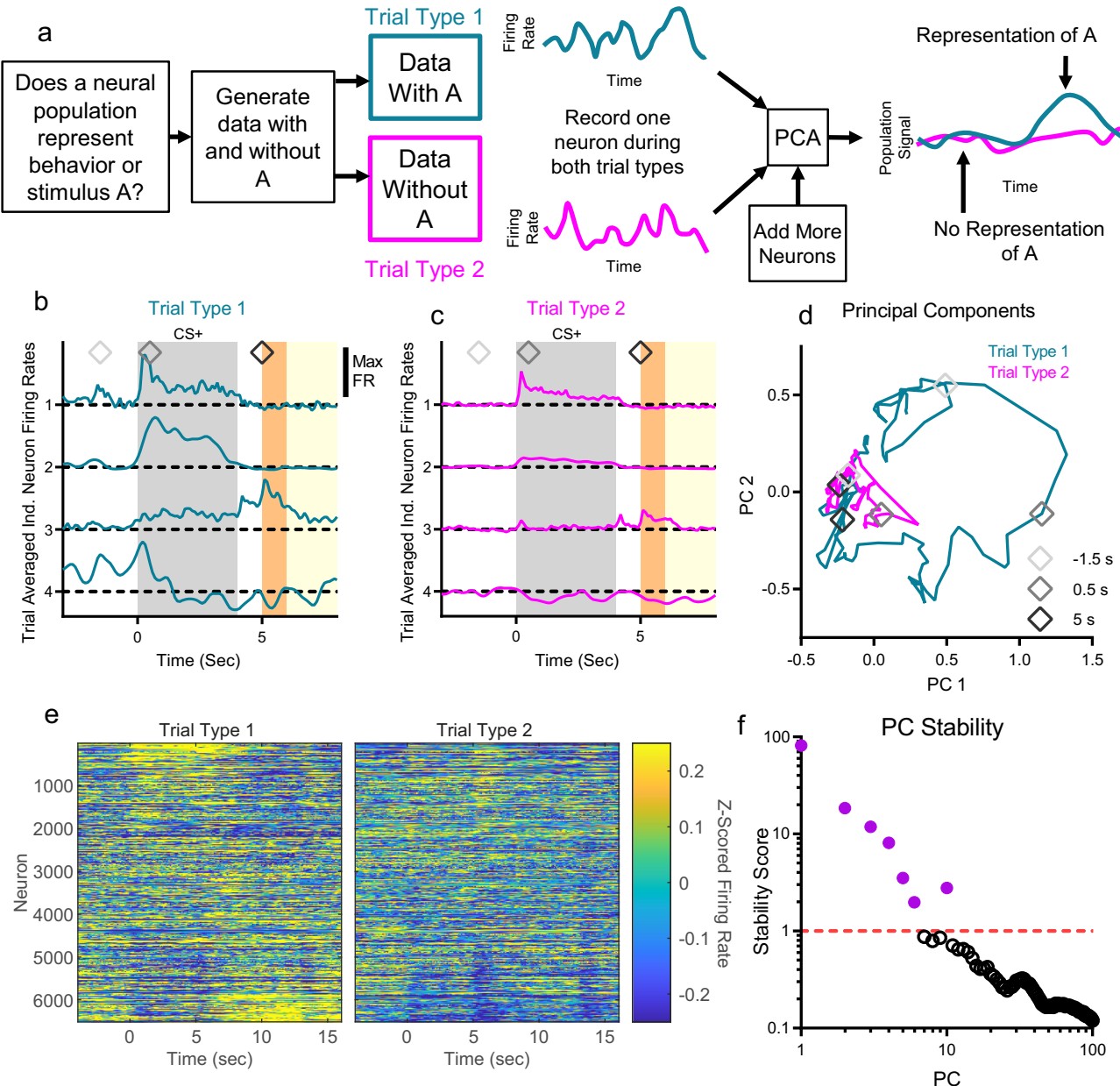

**Fig. 2 Neural population analysis using principal component analysis. a** We used principal component analysis (PCA) to assess neural representations of various decision-making relevant signals by comparing pairs of trial types that differentiate key stimuli (e.g., CS+ vs. CS-) or behaviors (e.g., seeking vs. not seeking, drinking vs. not drinking). **b, c** Four example neurons modulated their trial averaged firing rates dependent on trial type (baseline subtracted, firing rates scaled to maximum (bar), seeking vs. not-seeking state, see Fig. 3 for more information on seeking state). Trial type 1 consisted of data recorded when the animal was seeking alcohol and trial type 2 consisted of data recorded when the animal was not seeking alcohol (details of behavioral identification of seeking discussed below). Neurons 1 and 2 primarily responded to the CS+, though neuron 2 more clearly differentiated between trial type 1 and trial type 2. Neuron 3 responded to the CS+ and at the beginning of access on trial type 1. Neuron 4 differentiated trial types prior to the CS+ onset. **d** PCA reduced the four neuron firing rates to population signals (PCs) that are linear combinations of individual neuron firing patterns. The PC trajectory for trial type 1 is larger, indicating a more robust population signal. **e** Neurons sorted by PC linear combination coefficient (neuron identity matched in right and left panels). Firing responses before CS+ onset, during the CS+, and during access are visible (event timings same as b, but time frame longer in **e**). **f** Seven PCs were found to be stable across subsampling trials, which were used to control for differences in neuron yield across experimental groups and will be explored in the subsequent analyses. These selected PCs accounted for 38% of the overall variance in neural firing. Note that these stable PCs consisted of the first six PCs and PC 10.

rats (Wistars), but decreased it in compulsive rats (P rats) (Fig. 4d, strain main effect: F(1,38) = 80.6, $p = 6.2*10^{-11}$, time main effect: F(19,38) = 17.2, $p = 3.3*10^{-13}$, strain/liquid interaction: F(1,38) = 115, $p = 4.8*10^{-13}$).

Next, we compared firing patterns during the CS+ and CS- to examine representations of CS+/CS- discrimination. PC

trajectories for CS+ and CS- trials were similar following the initial CS onset, but CS- trajectories returned to the pre-CS position in activity space while CS+ trajectories did not (Fig. 4e, Supplementary Fig. 6). All groups exhibited increased PC separation during the CS+/- epoch (Fig. 4f). CS+/- representation strength (across the CS epoch) decreased only for compulsive

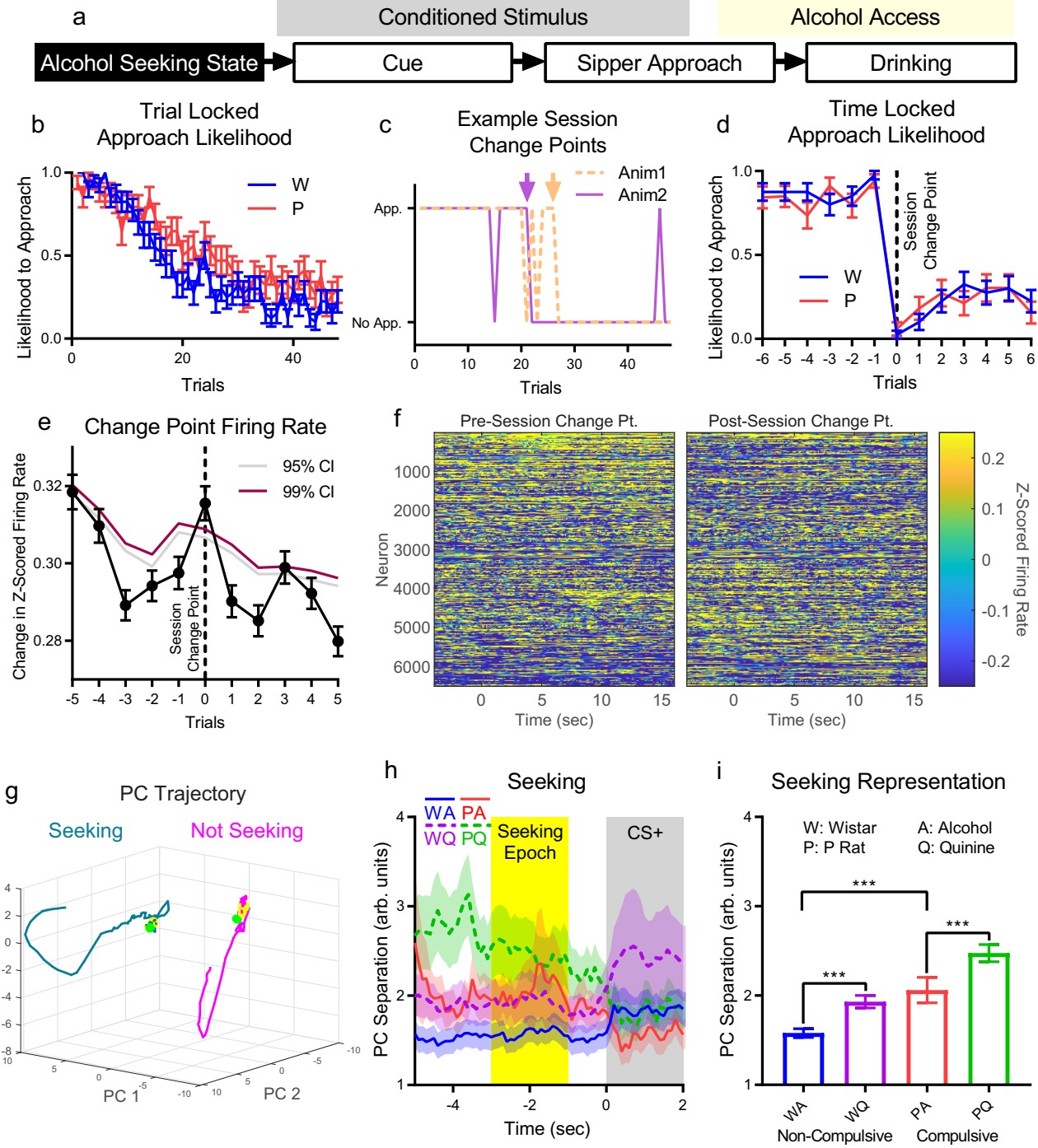

rats (P rats) in challenged drinking (Fig. 4g, liquid main effect: F(1,38) = 8.70, $p = 0.0054$, strain/liquid interaction: F(1,38) = 9.93, $p = 0.0032$, post-hoc Tukey's test, Wistar/Alcohol vs. Wistar/Alcohol+Quinine: $p = 0.99$, Wistar/Alcohol vs. P Rat/Alcohol: $p = 0.79$, P Rat/Alcohol vs. P Rat/Alcohol+Quinine: $p = 6.1*10^{-4}$). Overall, cue representation strength was similar in compulsive and non-compulsive rats, except for decreased representation strength in compulsive rats during challenged drinking.

**Approach initiation representation is enhanced in non-compulsive rats but diminished in compulsive rats during challenged drinking.** Sipper approach initiation was the first physical action in the consumption process (Fig. 5a). We identified the approach initiation time on a given trial as the largest

magnitude change point in the instantaneous likelihood to approach the sipper (Fig. 5b–d, see Methods). When neural firing patterns were time locked to the approach-initiation time point, we found a wide range of dynamic response patterns near the approach-initiation time point (Fig. 5e–g, Supplementary Fig. 7). PC trajectories for no-approach trials were relatively compact, whereas PC trajectories for approach trials showed movement through population activity space from an initial point several seconds before approach initiation (Fig. 5h). Despite the variety of dynamic patterns during approach initiation, the total PC separation for each experimental group was relatively stable through time (Fig. 5i). To assess the approach-initiation representation, a time window was selected −3 to −1 s before approach initiation (Fig. 5i, yellow window) to reduce any effects

**Fig. 3 Challenged drinking strengthens seeking representation in dmPFC in compulsive and non-compulsive rats. a** Alcohol seeking was assessed by examining differences in neural population firing patterns prior to task stimuli between approach and no approach trials. Approach was determined by movement to a sipper and into a position to lick from that sipper, as assessed by animal tracking. Animals rarely stayed at one sipper between trials (see sipper occupancy in Fig. 1f prior to CS) or failed to drink if they approached the correct sipper during access (95.08% of correct approaches resulted in at least one lick). When averaged throughout the session, likelihood to approach gradually decreased (**b**), but when time locked to the session change point for each animal, likelihood to approach exhibited a sharp change (**c** clear examples, arrows mark change point trials, **d** mean across sessions). Pre- and post-session change point trials were identified as seeking and not-seeking trials, respectively. **e** Change in average firing rate between trials increased at the session change point above randomized data (firing rate change jittered by up to +/− 2 trials). (**b**, **d**, and **e** mean +/− sem. b and d: N = 40 Wistar sessions from 7 animals, N = 34 P rat sessions from 7 animals. **e** N = 6514 neurons from 14 animals over 74 sessions) **f** After PCA and sorting neurons by their contribution to PC 1, a large remapping in active neurons was observed across trials immediately before and after the session change point (mean over last/first three trials shown). **g** Trajectories for seeking and not-seeking states were shifted in population activity space. (Mean trajectory, all neurons, only the 3 PCs with large separation effect size shown for clarity. −5 (green dot) to +7 s relative to CS+ onset. Yellow sections: seeking epoch (see **h**).) **h** PC separation between seeking and not-seeking trials across all seven stable PCs for each experimental group. (Mean +/− std across PCA subsample trials, only neurons from a given experimental group.) Seeking epoch was selected to capture representations before the CS+, but avoid overlapping in time with CS+/CS−. **i** Representation strength increased during challenge (quinine) for both strains and was higher in compulsive rats. (Mean +/− std shown across mean PC separation values from each 0.1 s time bin (N = 20) in seeking epoch (**h**, yellow region). ***: $p < 10^{-3}$.) Source data are provided as a Source Data file.

of movement initiation near the approach-initiation time point. The mean PC separation for each 100 ms time bin in this epoch was higher in non-compulsive than compulsive rats during alcohol-only drinking (similar to our previous work), and increased in non-compulsive rats, but decreased in compulsive rats during challenge drinking (Fig. 5j, strain main effect: $F(1,38) = 1332$, $p = 3.4*10^{-31}$, liquid main effect: $F(1,38) = 27.0$, $p = 7.2*10^{-6}$, strain/liquid interaction: $F(1,38) = 202$, $p = 8.8*10^{-17}$, post-hoc Tukey's test, Wistar/Alcohol vs. Wistar/Alcohol+Quinine: $p = 3.8*10^{-9}$, Wistar/Alcohol vs. P Rat/Alcohol: $p = 3.8*10^{-9}$, P Rat/Alcohol vs. P Rat/Alcohol+Quinine: $p = 1.05*10^{-6}$). Therefore, approach initiation representation strength was depressed in compulsive rats drinking alcohol, while challenged drinking increased representation strength in non-compulsive rats and depressed it yet further in compulsive rats.

**Drink representation is enhanced in non-compulsive rats during challenged drinking.** We examined the neural population representation of drinking by comparing CS+ drink and CS+ no drink trials (Fig. 6a). PC trajectories for no-drink trials were relatively compact while trajectories for drink trials showed large divergences in population activity space (Fig. 6b, Supplementary Fig. 10). PC separations (during the 2 s following drinking initiation) for non-compulsive rats during challenged drinking possessed a large peak immediately following the onset of drinking (Fig. 6c), indicating that dmPFC firing patterns in non-compulsive rats diverged substantially more when drinking alcohol + quinine when compared to other experimental groups, indicating a stronger representation (Fig. 6d, strain main effect: $F(1,38) = 155$, $p = 5.7*10^{-15}$, liquid main effect: $F(1,38) = 92$, $p = 1.1*10^{-11}$, strain/liquid interaction: $F(1,38) = 133$, $p = 5.6*10^{-14}$, post-hoc Tukey's test, Wistar/Alcohol vs. Wistar/Alcohol+Quinine: $p = 3.8*10^{-9}$, Wistar/Alcohol vs. P Rat/Alcohol: $p = 0.52$, P Rat/Alcohol vs. P Rat/Alcohol+Quinine: $p = 0.92$). This drink representation increase may underlie Wistar rejection of quinine-alcohol, while the lack of drink representation change may allow continued compulsive drinking in P rats.

**dmPFC excitation in P rats prevents the progression of compulsive drinking.** To better understand the large divergence in firing patterns in non-compulsive rats during challenged drinking (Fig. 6d), which coincided with marked reductions in alcohol consumption (Fig. 1j), we increased the data set with additional challenged drinking test sessions. Two later recordings (one from

a Wistar and one from a P rat) met the criteria to be switched between compulsive and non-compulsive groups due to reversed compulsive drinking behavior (Supplementary Fig. 14, see Methods for details). We examined drink representation and individual neuron firing responses during the drink epoch (Fig. 7a–f). Furthermore, we analyzed neuron waveforms to differentiate excitatory and inhibitory neurons[50,51] (Supplementary Fig. 11). Importantly, some types of inhibitory neurons have been shown to be indistinguishable from excitatory neurons based on waveform (e.g., in the amygdala[52,53]), so we refer to this classification as putative. In non-compulsive rats, challenged drinking continued to produce elevated drink representation and suppressed intake in later compulsive-drinking tests (Fig. 7a, Supplementary Fig. 12, test session main effect: $F(1,38) = 13.5$, $p = 7.3*10^{-4}$, liquid main effect: $F(1,38) = 244$, $p = 4.0*10^{-18}$, post-hoc Tukey's test, first Wistar/Alcohol vs. first Wistar/Alcohol+Quinine: $p = 3.8*10^{-9}$, first Wistar/Alcohol vs. later Wistar/Alcohol: $p = 0.44$, later Wistar/Alcohol vs. later Wistar/Alcohol+Quinine: $p = 3.8*10^{-9}$). The increase in drink representation during challenge (quinine-alcohol) corresponded to increased firing in both putative excitatory (liquid main effect: $F(1,3448) = 18.8$, $p = 1.5*10^{-5}$, post-hoc Tukey's test, first Wistar/Alcohol vs. first Wistar/Alcohol+Quinine: $p = 0.0015$, first Wistar/Alcohol vs. later Wistar/Alcohol: $p = 0.52$, later Wistar/Alcohol vs. later Wistar/Alcohol+Quinine: $p = 0.082$) and putative inhibitory (liquid main effect: $F(1,210) = 7.27$, $p = 0.0076$, post-hoc Tukey's test, first Wistar/Alcohol vs. first Wistar/Alcohol+Quinine: $p = 0.56$, first Wistar/Alcohol vs. later Wistar/Alcohol: $p = 0.99$, later Wistar/Alcohol vs. later Wistar/Alcohol+Quinine: $p = 0.018$) neurons (Fig. 7b, c). However, in compulsive rats, drink representation increased during additional drinking sessions regardless of liquid type and intake remained elevated (Fig. 7d, Supplementary Fig. 12, test session main effect: $F(1,38) = 161$, $p = 3.05*10^{-15}$, liquid main effect: $F(1,38) = 21.7$, $p = 3.8*10^{-5}$, time main effect: $F(19,38) = 2.09$, $p = 0.026$, test session/liquid interaction: $F(1,38) = 10.5$, $p = 0.0024$), which corresponded to increased firing in putative excitatory neurons only (Fig. 7f, test session main effect: $F(1,2523) = 66.88$, $p = 4.5*10^{-16}$, post-hoc Tukey's test, first P Rat/Alcohol vs. first P Rat/Alcohol+Quinine: $p = 0.38$, first P Rat/Alcohol vs. later P Rat/Alcohol: $p = 6.7*10^{-7}$, later P Rat/Alcohol vs. later P Rat/Alcohol+Quinine: $p = 0.82$). (See Supplementary Fig. 13 for inhibitory neuron firing time series.)

Since increased firing during challenged drinking was associated with reduced intake in Wistar rats, we hypothesized that dmPFC excitation suppresses compulsive drinking. Therefore, we

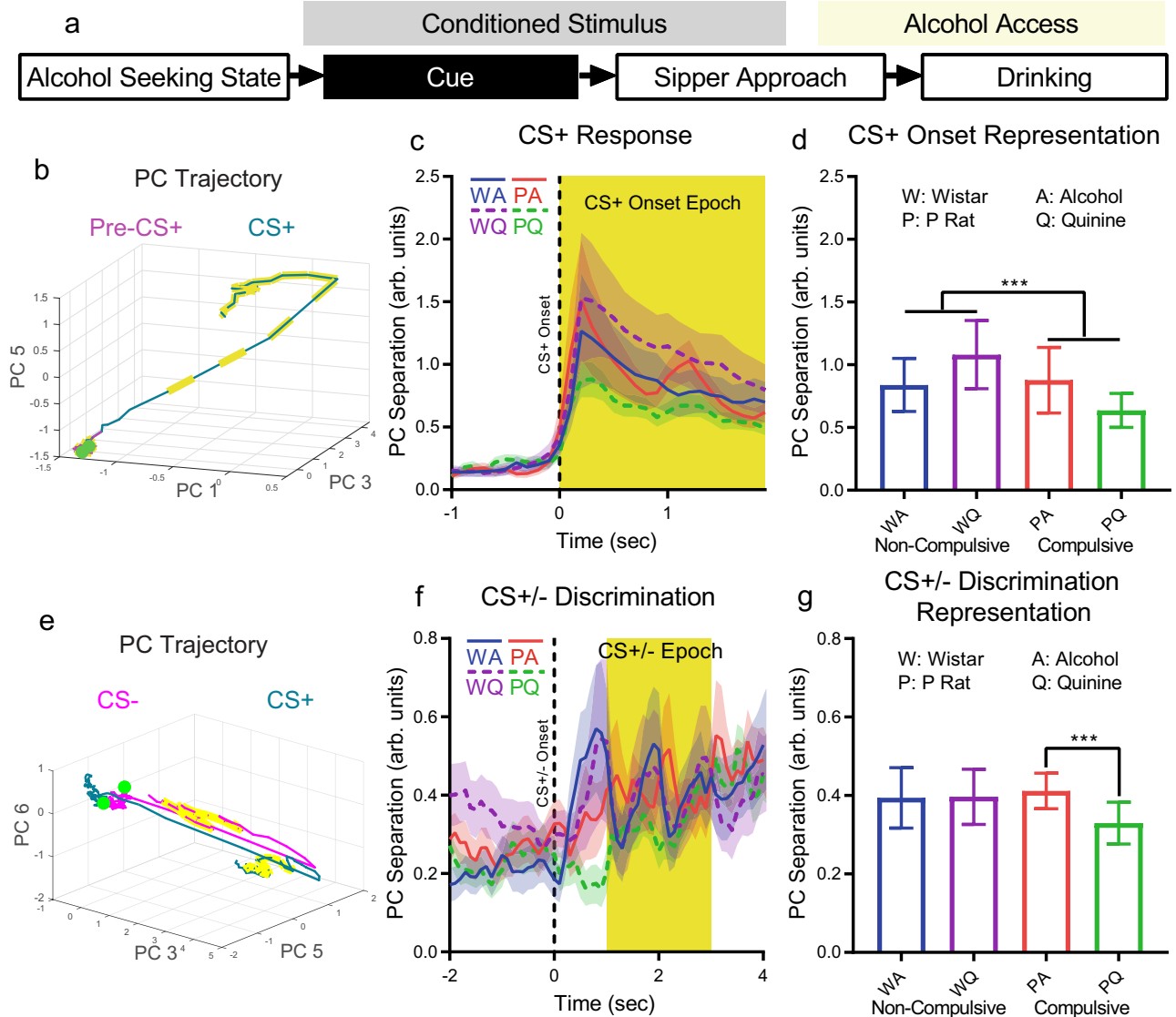

**Fig. 4 Challenged drinking improved or maintained cue representation in non-compulsive rats, but weakened cue representation strength in compulsive rats. a** Cue representation was examined by comparing pre-CS+ to CS+ time epochs (CS+ onset representation) and CS+ to CS− time epochs (CS+/− discrimination representation). **b** PC trajectories for pre-CS+ the time window did not vary for many PCs, consistent with a lack of stimulation or consistent time-locked behavior across trials. Conversely, CS+ onset produced a moving trajectory. (−1 (green dot) to +1.9 s relative to CS+ onset. Yellow dashed sections: CS+ onset epoch (see **c**).) **c** PC separation between pre-CS+ and CS+ time epochs. **d** Mean PC separation across the CS+ onset epoch (**c**, yellow epoch) showed that representation strength increased with challenged drinking (quinine) for non-compulsive rats, but decreased for compulsive rats. **e** Trajectories on both CS+ and CS− trials moved quickly from the pre-CS location near the start of the CS, but CS- trials quickly returned towards their original position. (−2 (green dot) to +4 s relative to CS+ onset. Yellow sections: CS+/− epoch (see **f**).) **f** PC separation between CS+ and CS− trials. Oscillations seemed to correspond to the on/off cycle of the blinking/solid light modality used for the CS+/− that was counter-balanced across animals. **g** Mean PC separation during the CS +/− epoch (**f** yellow time window) showed that CS+/− representation strength was unchanged in non-compulsive rats during challenge, but decreased in compulsive rats. **b**, **e**: mean trajectory, all neurons, only the 3 PCs with large separation effect size shown for clarity. **c** and **f**: mean +/− std across PCA subsample trials, only neurons from a given experimental group, all seven stable PCs. **d** and **g**: Mean +/− std across mean PC separation values from each 0.1 s time bin (N = 20) in CS+ onset or CS+/− epoch, ***: $p < 10^{-3}$. Source data are provided as a Source Data file.

conducted a second experiment in a new cohort of 20 P rats using identical alcohol exposure, 2CAP training, and compulsive drinking test procedures (Fig. 7g). We expressed Designer Receptors Exclusively Activated by Designer Drugs (DREADDs: AAV5-CaMKIIa-hM3D(Gq)-mCherry, control: AAV5-CaM-KIIa-mCherry) in dmPFC in this cohort of animals. During the first challenged drinking session with DREADD-mediated increased excitation following CNO injection (3 mg/kg), we found no differences between active DREADD animals and controls (t(18) = 0.25, p = 0.8), though we did find significantly

reduced consumption relative to baseline (one sample t-test) relative to $\mu_0 = 1$, t(19) = 10.53, $p = 2.3*10^{-9}$) (Fig. 7h). However, repeated CNO+quinine tests showed that consumption of alcohol+quinine increased in control animals (Fig. 7h, Supplementary Fig. 15, DREADD main effect across all 6 CNO+Qui tests: F(1,103) = 5.34, $p = 0.023$). CNO/DREADD did not alter locomotion (Supplementary Fig. 15). Thus, dmPFC excitation over time reduced the progression of compulsive drinking. Control tests performed between CNO+Qui tests 1 and 2 pointed to a possible explanation for the observed reduced consumption

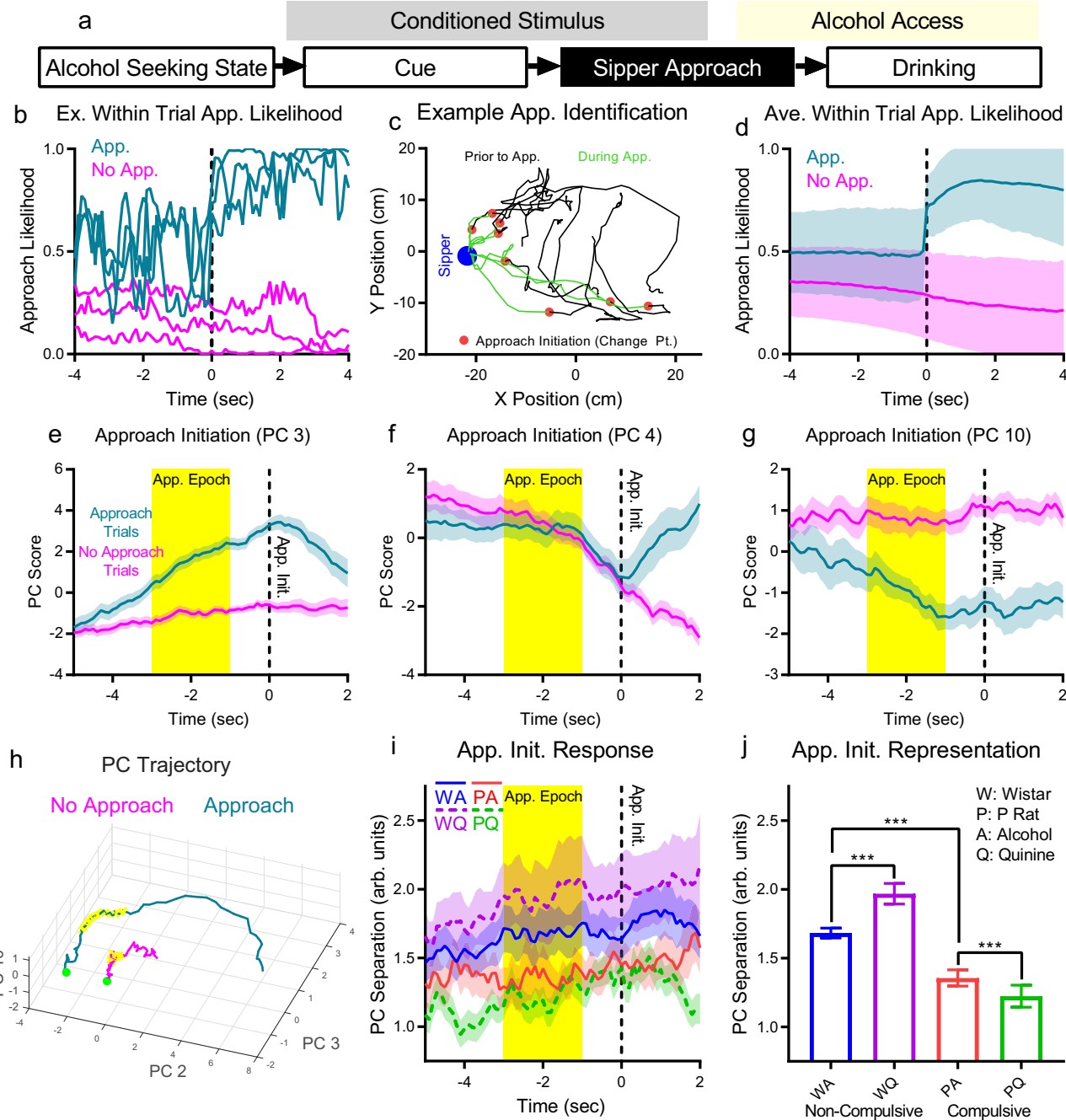

during CNO+Qui tests (Fig. 7i). These animals exhibited a smaller, though still significant, reduction in drinking during challenged drinking with quinine (compulsive drinking) and no injection (paired t-test with first CNO+Qui test, t(19) = 4.24, $p = 4.4*10^{-4}$, one sample t-test relative to $\mu_0 = 1$, t(19) = 3.22, $p = 0.005$), as well as when drinking alcohol during DREADD-mediated increased excitation following a CNO injection (paired t-test with first CNO+Qui test, t(18) = 4.24, $p = 4.9*10^{-4}$, one sample t-test relative to $\mu_0 = 1$, t(18) = 3.0, $p = 0.008$, no significant differences between active DREADD and control). However, injection of vehicle (i.e., no DREADD-mediated increase in excitation) paired with challenged drinking decreased consumption in both DREADD and control animals to a similar degree as was observed in CNO+Qui test 1 (paired t-test with first CNO+Qui test, t(19) = 0.96, $p = 0.35$), though not as substantially as was observed in Wistars (Fig. 1f, t-test with

Wistar relative consumption in Fig. 1f, t(24) = 5.1, $p = 3.2*10^{-5}$). These results indicate the existence of a synergistic effect between injection and quinine adulteration that reduced consumption.

## Discussion

We propose that challenged intake in rats demonstrating compulsive alcohol drinking was driven by a strong seeking representation and weak representations of behavioral control variables, such as cues, approach initiation, and drinking (Fig. 8). When grouped in this way, challenged drinking in non-compulsive rats resulted in increased engagement in both seeking state and behavioral control representations. Conversely, challenged drinking in compulsive rats resulted in a shift toward an enhanced seeking state representation and away from choice specific variable representations. (Note that, while drinking

**Fig. 5 Challenged drinking strengthened the representation of the sipper approach initiation signal in non-compulsive rats, but weakened the representation in compulsive rats. a** The representation of sipper approach initiation was examined by comparing approach to no-approach trials just before approach initiation. Change points in instantaneous likelihood to approach the sipper in each trial were used to identify sipper approach initiation (**b**: example approach likelihood traces on 6 trials, **c**: example animal position traces from eight approach trials marked with approach initiation, **d** average approach likelihood across all animals). On approach trials, the largest magnitude change point in approach likelihood marked the approach initiation. On no-approach trials, random approach initiation times relative to CS+ onset were used. (mean +/− std, $N = 1750$ approach trials and $N = 1802$ no approach trials from 14 animals across 74 sessions.) **e–g** PC scores on approach and no-approach trials began to diverge several seconds before approach initiation for PCs 3 and 10. Conversely, PC 4 only showed a divergence following approach initiation, indicating that this PC may track movement. (All PCs shown in Supplementary Fig. 8. **e–g**: mean +/− std across PCA subsample trials.) Though movement differences were found between approach and no-approach trials (Supplementary Fig. 8), these differences were stable through time prior to approach initiation, and correlations between PCs and animal speed were low (Supplementary Fig. 9), indicating that these PC results are not merely movement artifacts, though they may represent motor planning. **h** Large differences in PC trajectories were observed between approach and no-approach trials. (Mean trajectory, all neurons, only the 3 PCs with large separation effect size shown for clarity. −5 (green dot) to +2 s relative to approach initiation. Yellow sections: approach epoch (see **h**).) **i** PC separation between approach and no-approach trials all seven stable PCs. (Mean +/− std across PCA subsample trials, only neurons from a given experimental group.) **j** Mean PC separation across the pre-approach initiation epoch (**h**, yellow time window) showed that the representation was stronger in non-compulsive rats on alcohol days[20] and that representation strength increased with challenge (quinine) in non-compulsive rats, but decreased for compulsive rats. (Mean +/− std shown across mean PC separation values from each 0.1 s time bin ($N = 20$) in **f**. ***: $p < 10^{-3}$.) Source data are provided as a Source Data file.

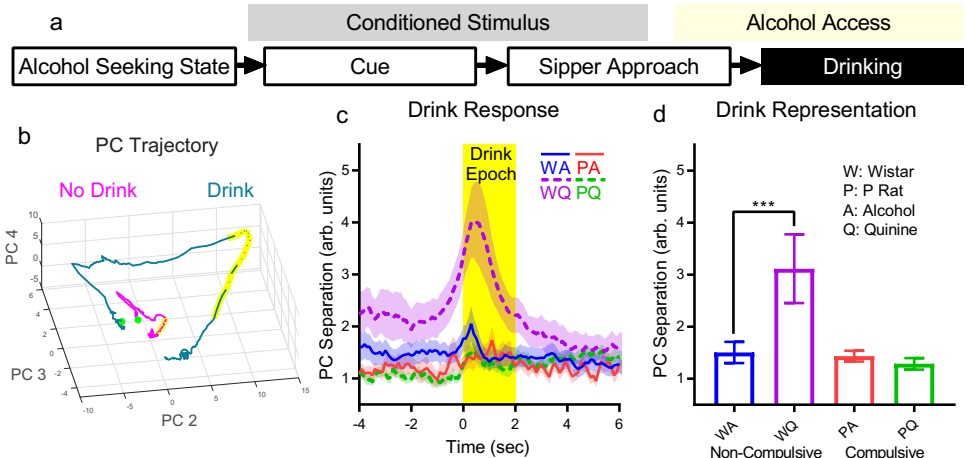

**Fig. 6 Challenged drinking strengthened dmPFC representation of drinking in non-compulsive rats, but not in compulsive rats. a** The representation of drinking was assessed by comparing neural firing patterns around the initiation of drinking, specifically with regards to differences between trials where the animal drank and trials where it did not. Drink initiation times were identified by tracking data and randomly selected drink initiation times relative to CS+ onset time were used as time lock points for no drink trials. **b** Drink trial trajectories showed a large loop in population activity space, whereas no drink trials were more compact. (Mean trajectory, all neurons, only the 3 PCs with large separation effect size shown for clarity. −10 (green dot) to +10 s relative to drink initiation. Yellow sections: drink epoch (see **c**).) **c** PC separation across all seven stable PCs during times immediately before and during drinking onset (Mean +/− std across PCA subsample trials, only neurons from a given experimental group). Non-compulsive rats during challenged drinking (quinine) showed a large peak in PC separation indicating a large change in population activity in response to quinine that was absent in other groups (i.e., a stronger representation of drinking). **d** Mean PC separation in the drink epoch (**c**, yellow time window) showed strengthened representations during challenged drinking (quinine) in non-compulsive rats. (Mean +/− std shown across mean PC separation values from each 0.1 s time bin ($N = 20$) in drink epoch. ***: $p < 10^{-3}$.) Source data are provided as a Source Data file.

representation increased during alcohol+quinine sessions in compulsive rats for later exposures, it was still similar to or lower than the drinking representation during later alcohol sessions.) Therefore, it may be possible to prevent compulsive drinking by increasing the representation of behavioral control variables in dmPFC during challenged drinking.

A striking result of these studies was the large increase in drink representation immediately after drinking onset in non-compulsive rats (Wistars) during challenged drinking. Furthermore, in these rats we found that excitatory and inhibitory neurons tended to increase their firing rate immediately after drink onset. This increase in drink representation strength and individual neuron firing response persisted through multiple compulsive drinking tests in non-compulsive rats. Conversely, compulsive rats (P rats) exhibited no change in drink representation or individual neuron firing rate during the first

compulsive drinking test. During later compulsive drinking tests, compulsive rats exhibited a large increase in drink representation and excitatory neuron firing during both unchallenged and challenged drinking sessions.

While these patterns of drink representation increases are similar, there are subtle differences in the conditions that produce these increases that point to possible explanations for dmPFC's role in compulsive drinking. Non-compulsive rats exhibited increases in drink representation only during challenged drinking, when their expectation about the contents of the solution was violated, leading to response inhibition and decreased consumption. This behavior is similar to previous reports that the anterior cingulate cortex (ACC, a region overlapping dmPFC[25]) responds to unexpected negative outcomes (so called error-related negativities)[54] and that these large scale signals are caused by altered neural activity across many neurons[55]. Furthermore, neurons in ACC have been shown to

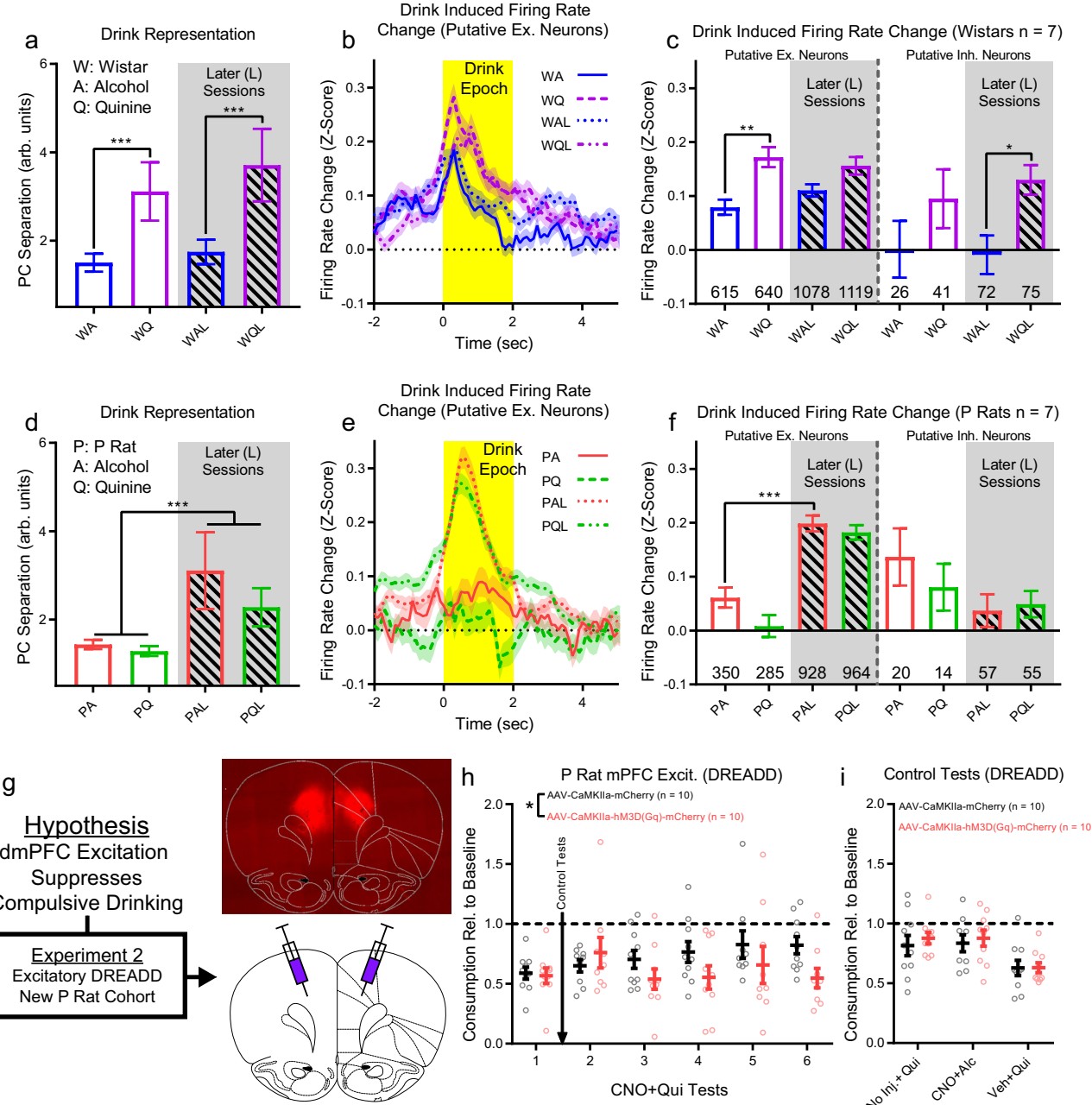

**Fig. 7 Changes in drink representation are related to putative excitatory/inhibitory neuron behavior and DREADD-mediated dmPFC excitation prevented the progression of compulsive drinking. a** Increased drinking representation strength in non-compulsive rats was persistent in later challenged drinking tests. This corresponded to increased putative excitatory (**b**, **c**) and inhibitory (**c** Supplementary Fig. 13) neuron firing. **d** Compulsive rats developed increased drinking representation during both challenged and unchallenged drinking, corresponding to increased firing in only putative excitatory neurons (**e**, **f**). **g** Thus, non-compulsive animals exhibited an increase in putative excitatory neuron firing on quinine-alcohol days relative to alcohol-only that was *not* observed in compulsive animals. Therefore, we hypothesized that driving dmPFC excitatory neurons would suppress compulsive drinking in compulsive animals. To test this hypothesis, a new cohort of 20 P rats underwent bilateral excitatory CAMKII-a DREADD or control virus injections in dmPFC (exemplary expression image, atlas image reproduced from ref. [24]). **h** After multiple challenged drinking tests, control animals exhibited increased compulsive drinking, while CNO/DREADD animals remained relative unchanged. **i** Animals expressed compulsive drinking, alcohol consumption was only slightly reduced by CNO, but consumption decreased for challenged drinking during vehicle injection, regardless of virus expressed. (**a** and **d**, mean +/− std shown across mean PC separation values from each 0.1 s time bin ($N = 20$) in drink epoch (first session results reproduced for comparison (Fig. 6d)). **b** and **e**, mean +/− sem change in firing rate on drink trials, see **c** and **f** for number of neurons. **c** and **f**, mean +/− sem of mean neuron firing rate change in drinking epoch, number of neurons shown in graph above horizontal axis. For all graphs, ***: $p < 10^{-3}$, **: $p < 10^{-2}$, *: $p < 0.05$.) Source data are provided as a Source Data file.

respond to conflict during response inhibition in stop-go or stop-change tasks[56,57]. Therefore, we hypothesize that increased drink representation during challenged drinking in non-compulsive rats was the neural signature of conflict processing in dmPFC that contributed to decreased consumption.

Compulsive rats exhibited increased drink representation during both challenged and unchallenged drinking in later sessions. A possible explanation for increased drink representation in later unchallenged drinking session is that the initial quinine challenge subsequently altered the expectation of the reinforcer,

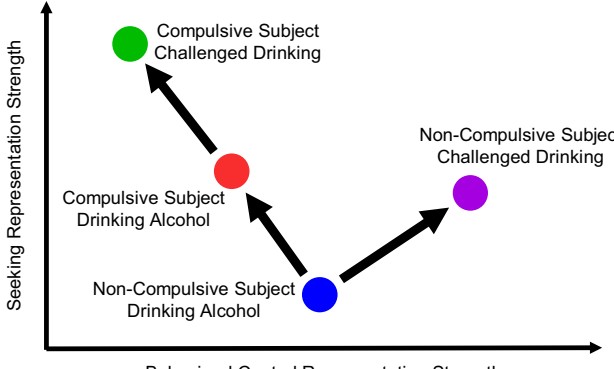

How do dmPFC representations of key alcohol consumption decision-making steps change with compulsive and challenged drinking?

**Fig. 8 Main conclusions.** Using electrophysiology in compulsive drinking (P rats) and non-compulsive drinking (Wistars) rats during their first challenged drinking test, we examined representations in dmPFC of seeking state and three behavioral control variables: cue, approach, and drinking. We labelled these as "behavioral control" variables because they represented the immediate, trial-by-trial signals that determined consumption on a given trial and because they changed in similar ways across groups. The seeking variable was different in that it operated on a longer time scale (i.e., across multiple trials). While the seeking variable was inferred behaviorally by actions governed by behavioral control variables (i.e., high likelihood to approach implied the animal was seeking), we conceptualized the seeking state of the animal as an initial condition that was antecedent to behavioral control. We found that dmPFC in non-compulsive rats more robustly represented behavioral control and seeking state signals when drinking was challenged by an aversive stimulus. For compulsive rats, relative to non-compulsive rats, dmPFC better represented alcohol seeking state, but represented a key behavioral control variable worse when drinking alcohol. When drinking was challenged in compulsive rats, the representation of behavioral control signals weakened and the representations of seeking strengthened yet further in dmPFC.

whereby the possibility of receiving unadulterated alcohol became less certain. We suggest that the increased drink representation during later challenged sessions relative to earlier sessions is a result of impaired conflict processing that results in a head-down-and-push behavioral strategy[17]. Importantly, non-compulsive animals exhibited an increase in the drink representation during challenged drinking relative to unchallenged drinking, which was absent in compulsive animals in the first and later sessions. The absence of this representation increase in compulsive rats could be due to many factors, including changes within dmPFC, changes in upstream brain regions that encode rewarding and aversive stimuli, and/or the connections from those upstream regions to dmPFC.

Based on the observed increase in excitation in non-compulsive rats during challenged drinking and transcranial magnetic stimulation studies in human substance use disorder patients[41,42], we hypothesized that exciting dmPFC in P rats would reduce compulsive drinking. To test this hypothesis, we expressed excitatory DREADDs in excitatory neurons (CAMKIIa promoter) in dmPFC in P rats. We found that activation of the DREADD did not eliminate but prevented the escalation of compulsive drinking. Therefore, we hypothesize that excitation of dmPFC excitatory neurons disrupted maladaptive processes necessary for the escalation of compulsive drinking. Indeed, recent research has shown that down regulation of *SYT1* (a gene that influences

synaptic transmission and plasticity) in prelimbic cortex increases compulsive alcohol consumption (assessed using quinine adulteration) and reduces neural excitability in Wistars with longer alcohol drinking histories[58]. Overall, while this and previous studies[21,22] have demonstrated that mPFC plays a functional role in compulsive drinking, this study also provided valuable insights into the neurocomputational phenomena underlying compulsive drinking. These results highlight the need for future experiments that simultaneously actuate and measure neural activity from neural ensembles of the dmPFC in animals drinking alcohol compulsively. These experiments will be capable of determining if the changes in computation described here are required for compulsive drinking.

With the strong correspondence of rat strain and the compulsive and non-compulsive phenotypes it's important to consider whether the differences we observed were merely strain differences between Wistars and P rats unrelated to compulsive drinking. While this possibility cannot be entirely eliminated, we believe this is unlikely for several reasons. First, we detected differences in the neural firing behavior of a brain region known to be involved in decision-making and substance use disorders, and we detected these differences immediately before and during the expression of compulsive drinking, making it unlikely that these differences were merely genotypic artifacts unrelated to compulsive drinking. Furthermore, some of these differences were observed during epochs where no behavioral differences were observed between strains (e.g., approaching) or during epochs where the neural signal differences connected logically to behavior (e.g., large drink response in Wistars during quinine drinking which corresponded to a halt in drinking), indicating that these differences were not merely behavioral artifacts unrelated to compulsive drinking. In addition, there are several compelling reasons to use these strains for comparisons involving compulsive drinking. P rats are an established pre-clinical rodent model of genetic risk for AUD[28]. Furthermore, differences between P rats and Wistars are likely related to alcohol consumption, given that P rats were bred from Wistars[29]. This common ancestry, along with the fact that Wistars can drink compulsively given sufficient drinking history[1,2,30–32], indicates that Wistars and P rats share many similarities and are well-suited comparisons.

It is possible that P rats are less sensitive to the taste of quinine in alcohol, which, if true, would likely influence the compulsive drinking patterns we observed. We showed that P rats find quinine aversive and prefer to drink alcohol that is unadulterated with quinine (see Supplementary Fig. 1), but it is possible that they find quinine less aversive than Wistars. Directly assessing this question would be difficult given confounds related to differences in alcohol consumption by P rats and Wistars, as well as confounds introduced by testing quinine sensitivity in different solutions. Furthermore, if it were true that P rats found quinine aversive, but not as aversive as Wistars, we would expect neural representation results for P rats drinking quinine to show similar, but weaker, changes relative to P rats drinking alcohol as were observed between Wistars drinking alcohol and alcohol+quinine. This was not what we observed. Often, the addition of quinine altered neural representations in P rats and Wistars independently of the patterns that were observed for alcohol alone.

Long drinking history Wistars[1,2,30–32] represent an alternative model for compulsive alcohol consumption that can provide valuable insights into the effects of alcohol exposure on compulsive drinking, as opposed to the genetic risk effects studied herein with P rats. One advantage to using P rats as a model of compulsive alcohol consumption is the greater similarity in alcohol exposure between compulsive and non-compulsive rats. Using long drinking history Wistars as a model of compulsive drinking

would involve greater differences in lifetime alcohol consumption and number of drinking sessions between compulsive and non-compulsive rats, which could complicate interpretation of results and change the focus of the study away from genetic risk. In the future, it would be interesting to see if long drinking history Wistars exhibit similar neurocomputational changes during compulsive drinking to those observed in P rats in this study.

## Methods

**Subjects**. This study was conducted using male alcohol preferring (P) rats (Indiana University School of Medicine) and male Wistar rats (Envigo, Indianapolis). All animals were shipped via ground transportation from breeding facilities to the laboratory, all within the city of Indianapolis. All animals were given food and water ad libitum in their home cages. All animal procedures were approved by the Indiana University – Purdue University Indianapolis School of Science Institutional Animal Care and Use Committee.

The electrophysiology study utilized 24 P rats and 72 Wistars over 4 sub-cohorts. Following IAP and 2CAP training, 8 P rats and 8 Wistars were selected for surgery (two sub-cohorts of 4 Wistars and two sub-cohorts of 4 P rats). One P rat and one Wistar were excluded from these sub-cohorts due to damaged probes, resulting in a final yield of 7 P rats and 7 Wistars. Electrophysiological data for one Wistar was lost for its first ARD test, so it was excluded from the analysis of the first ARD test. Later ARD test data were intact, so they were included in those analyses.

The DREADD experiment utilized 23 P rats. 20 animals were selected for virus surgery following IAP and 2CAP training (10 with DREADD, 10 Control). One DREADD animal was lost prior to the third from last test session.

**Intermittent Access Protocol (IAP)**. An intermittent access protocol (IAP) was used to acclimate animals to the taste and effects of alcohol[43,59]. All animals were 12 weeks old at the start of IAP. 20% ethanol (Decon Laboratories, Inc.) in one bottle and tap water in another bottle was provided ad libitum in the animals' home cages for 24 h periods on alternating days. Animals were weighed and bottles were placed on the cages on Monday, Wednesday, and Friday mornings (approximately 2 h into the animals' dark cycle) for 2 weeks. Bottles were pulled 24 h later and weighed to assess consumption. Animals were selected for 2CAP task training based on their average consumption across IAP (Supplementary Fig. 16). Within each sub-cohort of animals, we selected lower drinking P rats (with the exception of four P rats that did not drink) and higher drinking Wistars in an attempt to bring group consumption means closer together. We also removed animals with health concerns.

**Two-Way Conditioned Access Protocol Task (2CAP)**. The Two-Way Conditioned Access Protocol (2CAP)[20,43,44] task was used to assess voluntary consumption of 10% ethanol in tap water. All training sessions and DREADD testing sessions were conducted in standard rat shuttle chambers (MedAssociates) housed in custom sound attenuating boxes. Regular 2CAP sessions and baseline ARD testing sessions used 10% ethanol (Decon Laboratories, Inc.) in tap water. Quinine test sessions used a 10% ethanol solution adulterated with 0.1 g/L quinine (Fisher Scientific). Electrophysiological data were gathered in a custom made chamber to allow for sufficient space above the animal's head for the headstage and wiring. We utilized an updated version of the 2CAP task that included negative conditioned stimuli (CS−). Two possible stimuli modalities were used: a solid 4s light on stimulus or a 4s stimulus composed of four 300 ms light on followed by 700 ms light off blinks. Positive conditioned stimuli consisted of one modality of stimulus presented directly above the sipper where fluid would become available. Negative conditioned stimuli consisted of the other stimulus modality presented above both sippers. The specific modality type assignment (e.g., solid light CS+, blinking light CS−) was counter balanced across rats. The pseudorandom nature of the CS+ and CS− trials resulted in a slightly elevated likelihood (approximately 60% instead of 50%) for animals to receive a CS- trial following a CS+ trial and vice-versa. Each 2CAP session lasted approximately 1 h and consisted of 48 CS+ and 48 CS− trials. CS+ trials were separated by pseudorandomly selected intertrial intervals of 20, 28, 36, 44, 56, 68, 96, and 120 s. CS− trials randomly occurred within the intertrial intervals, though an exclusion period of 3 s at the start and end of CS+ trials prevented CS+ and CS− trials for occurring adjacent. Animals had a low likelihood to approach both sippers (i.e., approach the correct side after approaching the incorrect side) or approach the sipper prior to the end of the CS+ trial (Supplementary Fig. 17).

Animals were selected for surgery based on their average consumption across 2CAP training sessions (Supplementary Fig. 16). Within each sub-cohort of animals, we selected lower drinking P rats and higher drinking Wistars in an attempt to bring group consumption means closer together. We also removed animals with health concerns. Blood ethanol concentration (BEC) levels were gathered during a regular alcohol 2CAP session and during a free access session, but not during alcohol+quinine sessions in order to avoid impacting future alcohol+quinine drinking sessions. These BEC values were correlated with consumption (Supplementary Fig. 18).

**Electrophysiology**. Animals selected for electrophysiological recordings underwent implantation surgery using procedures approved by the Indiana University Animal Care and Use Committee. Animals were anesthetized using isoflurane and held in a stereotax (Kopf). The scalp of the animal was shaved and cleaned with alcohol and betadine. Cefazolin (30 mg/kg) was administered IP and Bupivacaine (5 mg) was injected subcutaneously in the scalp. After removing the scalp and clearing the bone, bregma and lambda were located. Grounding screws were implanted posterior to lambda above the cerebellum and 6 mounting screws were implanted (4 between lambda and bregma, 2 approximately 5 mm anterior of bregma). A trephine was used to perform a craniotomy above medial prefrontal cortex. For all animals except one P rat, a multishank probe (Cambridge Neuro-Tech, 64 channels on 4 or 6 shanks) was attached to a moveable drive and implanted with target coordinates of 3.2 mm A/P, 0.8 mm M/L, 3.0 mm D/V[24] in the right hemisphere with a 7.5 degree angle in towards the midline. The probe shanks were aligned so they spanned the anterior/posterior direction. One P rat was implanted with a fixed single shank probe (Cambridge NeuroTech, 32 channels) with target coordinates of 3.2 mm A/P, 0.6 mm M/L, 4.3 mm D/V[24] in the right hemisphere with a 7.5 degree angle in towards the midline. Movable drives were incased in antibiotic ointment and then affixed to the skull using dental cement. Ketofen (5 mg/kg) was administered and the animal was monitored for 7 days post-surgery.

Following recovery, rats were given two 2CAP sessions in a specialized electrophysiology chamber to reacclimate to the task and acclimate to the new chamber. Next, rats underwent at least 3 2CAP sessions with electrophysiology wires attached to acclimate to the presence of the wiring. Omnetics headstages were plugged in to the exposed connectors from the neural probes on the dental cement headcap. Electrophysiological data and task specific variables (e.g., stimuli) from the Med Associates controller (MedPC v4.1) were recorded using Open Ephys v0.4.0 data acquisition systems. Video of the animal was recorded from above the chamber using a Logitech webcam (Logitech Webcam Software v2.80.853.0). Licks were recorded with piezo microphones (Korg) attached to sipper tubes. Animal tracking was performed offline using DeepLabCut[60] v2. Based on the manual identification of drinking trials and the tracking information, the drinking position of each animal was identified and the drinking initiation time was set as the first time point when the animal's snout entered the drinking position (within 9 pixels (< ~1 cm) of the manually identified sipper location). Spike sorting was performed using Kilosort2[45]. Moveable probes were lowered by approximately 0.1 mm the day before each ARD test baseline session. Typically, the probes were lowered after the rat ran a regular 2CAP session on Monday to prepare for ARD baseline and test days on Tuesday and Wednesday.

Final positions of shanks were verified via histology (Supplementary Fig. 19) and quantified (QuickNII-WHSRat-v3)[61]. Probe tracks were identified via glial fibrillary acidic protein (GRAP) staining visualized by Alexa Flour 555 and nucleus DAPI staining. There was a slight, but significant difference in mean medial/lateral placement of 0.11 mm between P rats and Wistars. There was a larger, significant difference in mean anterior/posterior placement of 0.81 mm between P rats and Wistars. To ensure that differences in recording sites could not explain differences in neural encoding, the correlation between neuron location (defined as the location of the shank on which the neuron was recorded) and neuron principal component coefficients for each stable principal component (see below) was calculated. Across all combinations of PC and the spatial dimension, the $R^2$ values for these correlations were less than 0.0117, with most values less than 0.001. This indicates that these differences in recording location cannot explain the observed differences in neural encoding between P rats and Wistars. Recordings from 6 Wistars in the first ARD test yielded 655 neurons on alcohol days and 697 neurons on alcohol+ quinine days. Recordings from 7P rats in the first ARD test yielded 383 neurons on alcohol days and 304 neurons on alcohol+quinine days. Recordings from 7 Wistars for later ARD tests (14 total tests) yielded 1181 neurons on alcohol days and 1235 neurons on alcohol+quinine days. Recordings from 7P rats for later ARD tests (10 total tests) yielded 1015 neurons on alcohol days and 1044 neurons on alcohol+quinine days. No correlation was observed between neuron firing rate and amount of alcohol consumed[62].

Among the later compulsive drinking electrophysiology sessions, two rats were switched between groups based on alcohol+quinine consumption (see Supplementary Fig. 14). These recordings were identified using thresholds in z-scored alcohol+quinine consumption and percent decrease in consumption during the alcohol+quinine session relative to the immediately preceding alcohol session. The z-score alcohol+quinine consumption threshold was set at +/− 1.4 (z-score calculated for each strain separately). The percent decrease in consumption threshold was set at 50%. If a P rat session had an alcohol+quinine consumption z-score less than −1.4 and a percent decrease in consumption greater than 50%, it was reclassified as a non-compulsive session. If a Wistar session had an alcohol+quinine consumption z-score greater than 1.4 and a percent decrease in consumption less than 50%, it was reclassified as a compulsive session. These switched recordings produced drink firing rate response patterns that largely agreed with the results observed in the rest of the analysis: increased firing in non-compulsive rats and no changes in compulsive rats during quinine adulteration (see Supplementary Fig. 14d, e).

Neurons were classified as putative inhibitory or excitatory neurons using their largest amplitude (across all electrodes) mean waveform (Supplementary Fig. 11). The waveform was rescaled from −1 (maximum hyperpolarization amplitude) to 1

(maximum post-action potential depolarization peak). Similar to previously used methods, we classified the neurons using time delays associated the waveform depolarization. We used the time to 50% depolarization and the time to 95% depolarization as the time delays to characterize the speed of the depolarization. When these delays for each waveform were plotted, two clusters were apparent. Waveforms were manually clustered into putative inhibitory (fast depolarization), putative excitatory (slower depolarization), and outliers based on these time delays.

**DREADD experiments**. Animals selected for DREADD experiments underwent virus injection surgery using procedures approved by the Indiana University Animal Care and Use Committee. Animals were anesthetized using isoflurane and held in a stereotax (Kopf). The scalp of the animal was shaved and cleaned with alcohol and betadine. Cefazolin (30 mg/kg) was administered IP and Bupivacaine (5 mg) was injected subcutaneously in the scalp. After removing the scalp and clearing the bone, bregma was located. Bilateral craniotomies were performed above dmPFC. Bilateral injections of 0.65 uL at 0.2 uL/min of virus were performed using Hamilton syringes and Harvard Apparatus pumps using the target coordinates of 3.2 mm A/P, 0.8 mm M/L, 3.0 mm D/V[24] with a 20 degree angle in towards the midline. Syringes were left in place for 10 min following injection and then removed slowly over 5 min. Ketofen (5 mg/kg) was administered and the animal was monitored for 7 days post-surgery. Virus expression was verified in all animals by histology (Supplementary Fig. 20).

Animals receiving the excitatory DREADD were given AAV5-CaMKIIa-hM3D(Gq)-mCherry (AddGene) and control animals received AAV5-CaMKIIa-mCherry (UNC Vector Core). Tests in which the DREADD was activated were conducted using 3 mg/kg Clozapine N-oxide (CNO, Cayman Chemical Company Inc.) in 1% dimethylsulfoxide (DMSO) and saline IP injections approximately 30 min prior to session start.

DREADD experiments took place in the same Med Associates shuttle chambers used for 2CAP training. Lick data was used to identify errors in consumption measures from bottle leaks using the following procedure. For each animal, number of licks and consumption for all baseline and test sessions were linearly fit. Then, the residuals for each session were calculated. The residuals for all sessions for all animals were then compared and residuals above three standard deviations were excluded. This resulted in 4 out of 197 test measurements being excluded.

**Electrophysiology data analysis**. Data analysis of spike trains was conducted using the following steps (see Supplementary Fig. 21). The raw spike trains (30 kHz) were rebinned using 100 ms bins (10 Hz). Each spike train was then smoothed using an adaptive Gaussian kernel with a standard deviation of one-quarter of the mean interspike interval for that neuron. We will refer to the smoothed spike train of neuron $i$ at time bin $t$ as $x_i(t)$. Next, based on the four stages of the decision to consume alcohol that were discussed (seeking state, cues, initiating approach, and drinking), we extracted segments of the spike train to form trials of interest.

For seeking state analyses, we focused on pre- and post-session change point CS+ trials. Let $t_{CS+,j}$ represent the time bin $t$ of the $j$th CS+ trial relative to CS+ onset. We extracted segments of the spike trains near the CS+ ($-10$ s before CS+ onset to 20 s after CS+ onset). Let $a_{i,j}(t)$ represent the smoothed spike train of neuron $i$ at time bin $t \in [-100, 200]$ relative to $t_{CS+,j}$ on trial $j$. Next, we z-scored $a_{i,j}(t)$ across all time bins and trials. Then, we averaged across trials before the session change point trial ($j_{scp}$) to obtain a mean high seeking state trail ($a_i^+(t) = \langle a_{i,j}(t)\rangle_j$ such that $1 \le j < j_{scp}$) and we averaged across trials after the session change point trial to obtain a mean low seeking state trial ($a_i^-(t) = \langle a_{i,j}(t)\rangle_j$ such that $j_{scp} \le j \le j_{max}$). The session change point trial was identified by constructing a sequence of approach (both correct and incorrect sipper, coded as 1) and no-approach (coded as 0) in time order for each CS+ trial. This sequence was analyzed using the Matlab function findchangepts.m to find the largest magnitude change point.

For analyses of drinking cues, we focused on CS+ and CS- trials. Let $t_{CS,j}$ represent the time bin $t$ of the $j$th CS trial relative to CS onset. We extracted segments of the spike trains near the CS ($-10$ s before CS onset to 20 s after CS onset). Let $b_{i,j}(t)$ represent the smoothed spike train of neuron $i$ at time bin $t \in [-100, 200]$ relative to $t_{CS,j}$ on trial $j$. Next, we z-scored $b_{i,j}(t)$ across all time bins and trials. Then, we averaged across CS+ trials to obtain a mean CS+ trail ($b_i^+(t) = \langle b_{i,j}(t)\rangle_j$ such that $j$ is a CS+ trial) and CS- trials to obtain a mean CS- trial ($b_i^-(t) = \langle b_{i,j}(t)\rangle_j$ such $j$ is a CS- trial).

For approach initiation analyses, we focused on approach (both correct and incorrect sipper) and no-approach CS+ trials. Trials were marked as approach trials if the animal approached either sipper during the access period. The approach initiation time was identified using a change point analysis. Let $\vec{z}_i(t)$ represent the position and velocity of the animal's snout at time bin $t$ relative to the CS+ onset on trial $i$. Using both the position and velocity of the animal's snout allowed the algorithm to account for differences in movements to approach the sipper based on different initial positions and velocities at the start of the trial. Let $p_j$ represent the approach status for trial $j$ (1 for approach trials and 0 for no-approach trials). We calculated the likelihood that the animal would approach at time bin $t$ on trial $i$

($q_i(t)$) using the weighted average of the approach status ($q_i(t) = \sum_{j\ne i} w_{i,j}(t)p_j$) where the weights were the normalized inverse Euclidean distance between the locations and velocities of the animal's snout at the same time bin relative to CS+ onset on two trials (i.e., $\vec{z}_i(t)$ vs. $\vec{z}_j(t)$). This sequence was analyzed using the Matlab function findchangepts.m to identify the largest magnitude change point in the approach likelihood which represents the approach initiation time bin on trial $i$ ($t_{AI,i}$). For no-approach trials, the approach initiation time relative to the CS+ onset was randomly selected from the approach initiation times relative to the CS+ onset from approach trials for that animal on that day. We extracted segments of the spike trains near the approach initiation time ($-5$ s before approach initiation to 2 s after approach initiation). Let $c_{i,j}(t)$ represent the smoothed spike train of neuron $i$ at time bin $t \in [-50, 20]$ relative to $t_{AI,j}$ on trial $j$. Next, we z-scored $c_{i,j}(t)$ across all time bins and trials. Then, we averaged across approach trials to obtain a mean approach trail ($c_i^+(t) = \langle c_{i,j}(t)\rangle_j$ such that $j$ is an approach trial) and no-approach trials to obtain a mean no-approach trial ($c_i^-(t) = \langle c_{i,j}(t)\rangle_j$ such that $j$ is a no-approach trial).

For the analysis of drinking, we focused on correct approach trials and not correct approach trials (i.e., incorrect approach and no-approach trials). We observed that animals that correctly approached (i.e., were in the drinking position at the extended sipper during access) almost always licked the sipper at least once (95.08% of correct approach trials based on manual coding of video, audio, and piezo microphone data during the first baseline and quinine sessions), though we were unable to resolve the precise lick time. Therefore, we refer to correct approach trials as drink trials. Let $t_{D,j}$ represent the time bin of the $j$th CS+ trial when the animal first arrived at the sipper (i.e., drink time). For no drink trials (i.e., trials where the animal did not approach either sipper or approached the incorrect sipper), we randomly selected drink times relative to the CS+ onset from drink times relative to the CS+ onset on drink trials. We extracted segments of the spike trains near the drink time ($-10$ s before drink time to 10 s after drink time). Let $d_{i,j}(t)$ represent the smoothed spike train of neuron $i$ at time bin $t \in [-100, 100]$ relative to $t_{D,j}$ on trial $j$. Next, we z-scored $d_{i,j}(t)$ across all time bins and trials. Then, we averaged across drink trials to obtain a mean drink trail ($d_i^+(t) = \langle d_{i,j}(t)\rangle_j$ such that $j$ is a drink trial) and no drink trials to obtain a mean no drink trial ($d_i^-(t) = \langle d_{i,j}(t)\rangle_j$ such that $j$ is a no drink trial).

All trial means of interest were concatenated through time for each neuron (i.e., the sequence $A_i(t) = [a_i^+(t), a_i^-(t), b_i^+(t), b_i^-(t), c_i^+(t), c_i^-(t), d_i^+(t), d_i^-(t)]$) and principal component analysis (PCA) was performed on 500 iterations of 200 randomly selected neurons from each group (animal strain, liquid type (alcohol vs. alcohol+quinine), and ARD test number (first test vs. later tests)). We ran the PCA with neurons considered as variables and time bins considered as observations in order to capture patterns of neural activity across neurons at each time bin of the trials of interest. This process of randomly subsampling neurons and matching the number of neurons from each experimental group in the PCA prevented one group from dominating the PCA, but allowed the PCA to be meaningful across groups.

For each principal component (PC), we determined the stability of the PC across the 500 iterations using the following method. Let $e_{i,j}(t)$ represent principal component $i$ at time bin $t$ in the concatenated sequence of trials on PCA iteration $j$. PCs that were found to be inverses of the mean PC across all iterations were inverted by multiplication with $-1$. For each PC, we calculated the ratio of the variance in the mean PC throughout time to the variance across iterations using $var_t\left(\langle e_{i,j}(t)\rangle_j\right)/var_t\left(e_{i,j}(t) - \langle e_{i,j}(t)\rangle_j\right)$. If the variance ratio was above 1, we considered the PC to be stable across PCA iterations and used it in subsequent analyses.

PCs for each experimental group (e.g., Wistars drinking alcohol+quinine in the first compulsive drinking test) were generated by projecting from only the 200 neurons from that group that were used in each of the 500 iterations. Let $f_{i,j,k}$ represent the coefficient for neuron $i$ for PC $j$ on PCA iteration $k$. The projection for PC $j$ on PCA iteration $k$ for a given experimental group were calculated as $E_{j,k}(t) = \sum_{i\in G} f_{i,j,k} A_i(t)$ such that $G$ is the set of 200 neurons from the experimental group of interest that were randomly selected for this PCA iteration. Unless otherwise stated, the mean across 500 iterations was calculated for PC projections by experimental groups and for all neurons together. PC separations at each time bin were calculated as the Euclidean distance between all stable mean PCs for the two types of interest (e.g., high seeking vs. low seeking) at corresponding time points in the trials (e.g., 2 s before CS+ onset).

A leave one out analysis was performed by rerunning the analysis with every combination of one animal removed. Only minor qualitative differences were observed. Furthermore, a randomized analysis was performed by shuffling assignments for the trial types of interest (e.g., before and after session change point when assessing seeking state). PC stability and PC separations were substantially reduced, indicating that the signals observed herein represent true neural population signals associated with the decision-making signals of interest.

Neural encoding/decoding analyses are a related method to the PCA method we utilized in this analysis. In these encoding/decoding analyses, the experimenter quantifies how well neural signals encode various stimuli or behaviors, or how well the stimuli or behavior can be decoded from the neural signal. In other words,

these analyses ask how separable the neural signals are during stimulus A trials versus stimulus B trials, for instance. In contrast, we quantified how different the neural signals were during contrasting stimuli or behaviors by examining PCs. We performed a neural decoding/encoding analysis[63] using the same underlying data structures that were used for the PCA (Supplementary Figure 22). When we examined encoding/decoding by individual neurons, we found few neurons (0–5% in most cases) significantly encoded the decision-making relevant variables that were the focus of this manuscript. This result further motivates the importance of examining neural population signals. When we examined encoding/decoding by PCs, we found that many PCs encoded the decision-making relevant variables to a high degree, in agreement with large and stable separations in PC traces observed throughout trials and epochs of interest (see Supplementary Figs. 3, 5–7, and 10).

**Statistics**. For comparisons between P rats and Wistars across alcohol sessions and alcohol+ quinine sessions, we utilized a two-way ANOVA to detect main effects of rat strain and liquid type, as well as an interaction. We used the same procedures for other two-way comparisons, such as first compulsive drinking test sessions vs. later sessions and alcohol vs. alcohol+quinine sessions. For comparisons between P rats and Wistars across alcohol and alcohol+ quinine sessions that also incorporated measurements through time (e.g., comparisons of PC separations during epochs of interests), we utilized a three-way ANOVA to detect main effects of rat strain, liquid type, and time, as well as strain*liquid and strain*liquid*time interactions. In the main text, we report only significant ($p < 0.05$) results of these comparisons. We followed these ANOVAs with post-hoc Bonferroni corrected two-sided t-Tests (behavior, Fig. 1) or Tukey–Kramer tests (all other figures) and we report the results for three comparisons of interest: Wistars/alcohol vs. Wistars/alcohol+ quinine, P Rats/alcohol vs. P Rats/alcohol+ quinine, and Wistars/alcohol vs. P Rats/alcohol. Significant results of these post-hoc comparisons are reported in figures only. These post-hoc comparisons were not performed or reported in the case of a significant main effect of time in three-way ANOVAs and relevant other significant main effects are marked in figures instead. We used the same two-way approaches for the analyses of DREADD vs. control animals and multiple drinking sessions. Other two-sided t-tests were used in behavioral analyses (Figs. 1 and 7). Statistical analyses were performed in Matlab (R2020a) and Prism (v7.05).

Throughout figures, error bars and fringe represent standard error of the mean (sem), with the exception of PC separation plots and their summary bar graphs where we show standard deviation (std). We chose to show standard deviation for these graphs because the sample size was controlled by the number of subsamples used in the PCA method in those cases. In all figures, we identify in the caption the type of method used for each error bar or fringe to avoid confusion.

**Reporting summary**. Further information on research design is available in the Nature Research Reporting Summary linked to this article.

## Data availability
All spike sorted data and behavioral data utilized in this study are freely available in a figshare repository[64] [https://doi.org/10.6084/m9.figshare.19387511.v2]. Source data are provided with this paper.

## Code availability
All custom analysis software utilized in this study is freely available in a figshare repository[64] [https://doi.org/10.6084/m9.figshare.19387511.v2] with the data.

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

## Acknowledgements

This work was supported by NIH Grants: AA023786 (CCL), AA007462 (NT), AA022268 (DL), AA022821 (CCL), AA028265 (NT).

## Author contributions

N.M.T., D.L., and C.C.L. contributed to the conception and design of the study. N.M.T., B.M., E.C., and T.G. contributed to the acquisition of data. NMT conducted the data analysis and created software used in the analysis. N.M.T., D.L., and C.C.L. drafted the manuscript.

## Competing interests

The authors declare no competing interests.
