## [Peer Review File · Nature Communications]

Compulsive alcohol drinking in rodents is associated with altered representations of behavioral control and seeking in dorsal medial prefrontal cortexREVIEWER COMMENTS

Reviewer #1 (Remarks to the Author):

Timme et al, in their manuscript “Compulsive drinking is associated with neural activity patterns reflecting diminished behavioral control and enhanced seeking representations in dorsal medial prefrontal cortex”, examine how cortical populations differentially represent behaviors with respect to alcohol drinking. One purported phenotype of AUD is the continued consumption of alcohol in spite of associated negative consequences; and the hypothesis is that this is due to alterations in PFC circuits supporting decision-making. Understanding how these behavior-related computations have changes will shed light on alterations to mPFC contributions, and is an important question. Further, animals with different behavioral phenotypes are likely relying on different information to control behavioral output, so identifying which computations are different is important in order to assess their contribution to continued use.

Two strains of rats were exposed to initial alcohol consumption in their home cages, and the trained on a cued- alcohol availability task where CS+ predicted the soon-insertion of a alcohol sipper tube and CS- predicted absence. Animal tracking through deep lab cuts was used to nicely track animal position in the chamber. The authors were then able to use position in their analyses terming time spent by sipper outside CS+ and taking periods as “alcohol seeking”, akin to this reviewer as a modified conditioned place preference. Behaviors were quantified, and recording performed in mPFC populations of these two strains during 1) alcohol sessions and 2) alcohol + quinine sessions. They then showed that medial prefrontal cortex population activity correlates mapped onto principal component space showed different magnitudes of separation in relation to rat strain and challenged drinking. The majority of the manuscript is spent dissecting principal component separation in relation to a variety of behavioral variables, such as “seeking” behavior prior to the task cue. They show through primarily computational methodology how quinine drinking in non-compulsive animals is correlated with increased “engagement”, as quantified by principal component separation of mPFC population activity comparisons between trial types, in both seeking state and behavioral control representations. In contrast, quinine drinking in compulsive animals was correlated with greater “seeking representations”. Finally, they used a chemogenetic approach to prevent compulsive drinking observed in P-rats by increasing activity of mPFC to mimic the increased firing rate changes observed in non-compulsive animals during challenged drinking.

Overall, I am quite mixed, leaning more positive, on this manuscript. First, this is a beast of a paper, with a tremendous amount of interesting data. The authors should be commended for the computation approach taken; and indeed there is a lot of neural data presented in this paper (sometimes making it challenging to go through). The question asked is of interest, and the analyses showing the emergence of different representations (and how easily they were discriminable) were well-done. While I have concerns, they lay largely in more the interpretation and significance of the finding realm and I would like to hear what the authors response to the following is; 1) the determination of compulsivity is by strain, which raises the possibility that differences in representation differentiation may be a strain difference and not related to the “compulsive phenotype” as 2) very limited functional interrogation was performed to examine whether the emergence of these representations contributed to the ongoing behaviors (which would necessary to shown that seeking representations held within mPFC support compulsive

seeking. 3) While the PCA analyses are well-done and appreciated, I have some confusion given their selective presentation and the use of differentiation versus strength of contribution to behavior (no decoding). Point 2 may be beyond the scope of this already extensive body of work, but should be considered in the interpretation of what this is showing.

-Not clear is the justification for using PCA approach to examine activity differences between the two trial types as well as the two rat strains. While the authors address that they are not approximating strength of representations using trial classification methods, they do not make a strong case as to why they chose to use principal component analysis to quantify it instead. It is also unclear whether their main metric, "PC separation", is indicative of a "strength" of representation, rather than a selectivity for those populations to respective trial types. In several figures, it is unclear whether their findings are contingent upon arbitrarily selected principal components, rather than those which consistently explain the majority of the variance in their data (e.g. top 3 principal components). While the authors make sure to demonstrate that 7 of the principal components are represented in a stable manner across their captured population, it is also unclear why they chose to include PC 10 in a later analysis. Finally, it is not fully clear what their PC analysis adds towards measuring "representation strength" that straightforward comparisons of peri-event population activity would not. The richness of mPFC population activity changes across their behavioral variables could be similarly, and more simply, shown through peri-event activity comparisons (as seen in Fig 2B-C, E).

- It is hard to determine how much of the findings is due to strain comparison versus the emergence of a compulsive phenotype of mPFC representations. Classification of compulsivity is based on genotype (P rats bred to excessively drink alcohol)- with the compulsive behavior suggested as an endophenotype. Compulsive drinking is measured as the continued consumption of quinine-adulterated alcohol. Wistar rats (drink less alcohol at baseline) were used for comparison as a group who is more sensitive to quinine adulteration. So, in essence this is a strain comparison, and consumption led to blood ethanol concentrations that ranged between quite low, to slightly above the legal limit. Not clear to this reviewer is whether representation of task information is different between the two independent of a) the prior alcohol consumption history and b) use of alcohol as the reinforcer. Is that the different genotypes use different types of information during performance of the task (i.e. strain effect)? Is the argument that that emergence of the compulsive phenotype is going to coincide with emergence of these patterns of representations? There were no specific perturbations to mPFC activity during computation-specific epochs to assess function of these representational patterns to the support of the ongoing behavior. Does inhibition of mPFC activity in Wistar rats increase the consumption of quinine? Why use the two strains in this way?

- In relation to the above; all initial analyses were based on genotype- and not defined by a particular phenotype; however, two subjects were switched between groups based on phenotype? This is problematic, and one criteria should be used for group inclusion.

- Statistics do not always support the findings. For example, in Fig. 1e, examining consumption of quinine alcohol solution, there is a main effect of strain, and a main effect of quinine, but no interaction. So- overall, P rats drank more than Wistars, and overall, there was less quinine solution consumed, but not a difference in patterns of consumption suggested, "P rats showed a smaller reduction in consumption" is not supported. The claims or statements made in much of

paragraph 125-142 are not supported statistically.

- Further, for the DREADD experiments, the supporting statistics are very unclear. The first test should similar sensitivity to quinine and maybe a reduction from baseline (all determined visually)- then control tests were done showing variability in the quinine adulteration effect. Then, more CNO and quinine tests were performed. What is reported statistically for all of this is a there is a main effect of DREADD group; there seems to be many things going on in this within subject design. One is the separation of statistical analyses for the first test should be justified; second, the differences in the control tests need to be examined, and lastly, the continued test analysis does not support a pattern difference but an overall main effect of CNO. However, all of the above are just reporting group differences, with no analyses performed against baseline (1.0) (do they actually reduce drinking). There are group differences as well as magnitude effects on sensitivity to quinine (which the P rats seem to reduce their drinking by 50 %- not an insignificant amount and questions the assignment of the compulsivity label).

- More of the data used to support the approach to sipper as seeking should be reported. That animals rarely stayed at one sipper between trials or failed to drink during access has no data associated with it. Please provide data supporting these claims.

- Change point analyses: Interesting approach, more data is needed in the average number of change points exhibited- or the density of change point occurrences across the session. Did the authors just go with the first one? Looking at example Fig. 3c, it seems rats could have multiple change points? Given the alignment of neural activity, these details would be informative.

- Wave-form identification of cortical neuron type can be problematic; please use putative; not sure that such identification is necessary

- Is there concern that the use of DMSO, that produces long lasting smell/perhaps taste properties interfered with sensory (smell/taste) aspects contributing to alcohol consumption?

- 53 - can authors clarify what role they are referring to regarding mPFC and compulsive drinking?

- 67 - clarify what is meant by "some drinking sessions" in relation to experimental design

- could benefit from short discussion of what PCA approach could offer "regarding large-scale patterns of neural activity evolve during the compulsive decision to drink" that perievent activity does not.

- 149-152 - clarify epochs were defined by observer (e.g camera)

- 159-161 "Note that we do not attempt to identify which trial represents the signal, rather we only quantify separation as a measure of representation strength." what is the argument for not quantifying representation strength via trial identification (e.g. decoding analysis)?

- Fig 2b/c add y-scale or something, what do reductions mean?

- While the authors control for neuron yield across experimental groups, it would be of interest to see if differences observed can be explained by animal preference/aversion to the stimuli, e.g. do a couple of animals whose neurons have strong selectivity for alcohol/quinine drive the observed distances differences between population trial type PCs?

- Fig 3E unclear what the randomized data is on panel

- 253 attempt at explaining how variance explained by data in corresponding PC space is indicative of "strength" of a representation (but why not just compare firing rates of that neuron to the to trial types? what does PCA add?)

- Note: comparisons of PC separation within epochs could be confirmed further by using permutation tests between the two traces (e.g. Fig 3H,I), may even serve to show exactly when in these windows these differences exist

- Note: Confused as to why PC comparisons were made between different PCs (e.g. PC 1vs2vs4 in Fig 3, PC 1vs3vs5 in Fig 4 etc), do PC separation analysis rely on these hand-picked PC comparisons? Why not keep analysis consistent to, say, the top 3 PCs? It's confusing because if I compare PC 1 (e.g. 10% variance explained) to PC 6 (e.g. 1% variance explained) then it is not surprising to see these difference, hence why justification for PCs shown/analysis may be required. In the text, or figure legend, it is unclear if PC separation is being made for the PCs show in the 3D trajectory or for all 7 shown to be stable.

- Fig 5 Why bring PC 10 into analysis if only the first 7 were identified as stable?

- Figure 5 E-G Y axis meant to show PC score? if so, label appropriately and include in legend, not clear what is on the y scale

Reviewer #2 (Remarks to the Author):

In this study, Timme and colleagues used novel principal components and change point analyses to evaluate neural population representations in the dorsomedial prefrontal cortex (dmPFC) associated with compulsive alcohol drinking in rats. Results revealed several key differences in neural representations for behavioral control and alcohol consumption that could explain differences between compulsive and non-compulsive drinking rats. For instance, neural representations for alcohol seeking states were enhanced in compulsive drinkers. On the other hand, neural representations for cues associated to alcohol availability, as well as for approach initiation to the alcohol sipper and during actual drinking periods were diminished in the compulsive drinkers. In addition to these findings on neural representations, Timme and colleagues tested the idea that changes in neural activity in the dmPFC using chemogenetic-mediated manipulations could alter escalation of alcohol drinking in the compulsive rats. They found that chemogenetic-based excitation of dmPFC activity prevented escalation of compulsive drinking. Collectively, these findings support that notion that differences in neural activity profiles in the dmPFC underlie compulsive alcohol seeking and drinking.

Overall, I feel enthusiastic with the prospect of publication for this article. The text is very well written and organized, experiments are well controlled, and the data and results are very informative and well illustrated. Though, in the list below I highlight a few issues that require revision for improvement of the article and to further strengthen the main conclusions.

1. In Fig 7 (panels c and f), analyses on spike waveforms were performed to differentiate between excitatory versus inhibitory neurons. While this method used to be quite acceptable in the past, more recent evidence suggests that spike widths are not necessarily an accurate

method to tell excitatory versus inhibitory cells apart. For instance, while PV+ interneurons could have significantly shorter spike widths, SOM+ interneurons could have wider spike widths similar to principal neurons (Wolff et al., 2014; <https://pubmed.ncbi.nlm.nih.gov/24814341/>). In addition, some subpopulations of CCK+ interneurons also show wide spike widths (Vogel et al., 2016; <https://pubmed.ncbi.nlm.nih.gov/27497223/>). Thus, to avoid confusion or potentially misleading interpretations, I suggest moving this analysis on “excitatory” versus “inhibitory” neurons to supplemental materials. In addition, the authors should consider using the terms “putative excitatory” versus “putative inhibitory” neurons.

2. The rationale for the chemogenetic experiment in Fig 7g-i (lines 418-421) makes much sense in terms of the findings obtained from electrophysiology. In addition, the authors obtained findings that are consistent with the rationale (i.e., chemogenetic-mediated excitation of dmPFC prevented escalation of compulsive drinking in the P-rats). Though, the main conclusions of this article would get strengthened even further, perhaps to an almost irrefutable point, if the authors also perform the opposite experiment, i.e., chemogenetic-mediated inhibition of dmPFC in the Wistar rats to artificially induce compulsive drinking.

3. In Supp-Fig 19, it can be appreciated that some electrode placement tracks reached deeper areas considered as ventral mPFC. Where neurons from these excluded?

Anthony Burgos-Robles, PhD
University of Texas at San Antonio

Reviewer #3 (Remarks to the Author):

The report by Timme and colleagues describes the role of the dorsal medial prefrontal cortex (dmPFC) in controlling aversive-resistant drinking in a strain of rats selectively bred for high-alcohol preference. They use an interesting PCA-based method to study neural representation and report that chemogenetic activation of the dmPFC prevented the emergence of compulsive-like drinking. While the study is interesting, there are some major concerns that reduce enthusiasm for the manuscript. The main finding indicating a role of dmPFC in aversive-resistant drinking is also not entirely novel.

The validity of the compulsivity model presented by the authors is a major concern:

1) The numbers of animals used for the main behavioral experiment (Figure 1) is very low (not clearly indicated in the manuscript, but visual inspection of the figure seems to indicate a n=6 or 7 per group), ultimately leading to a low power for this analysis. This experiment requires more subjects to strengthen the conclusions.

Also, only one session was performed to assess baseline drinking levels, as well as the compulsive-like drinking using quinine adulteration. This is insufficient to confirm that a stable behavior was established and can be trusted to guide the following molecular investigations performed by the authors. Even more for the quinine adulteration experiment, drinking during the first day of exposure is very sensitive to the immediate aversive taste of quinine and can differ greatly from the stabilized drinking levels after at least a few sessions have been performed, in particular for Wistar rats.

2) The definition of the compulsive and non-compulsive groups is also an important issue. There are no clear criteria on how the authors classify compulsive vs non-compulsive animals, except the fact that there was a main effect of strain and solutions in Figure 1. The interaction between the two factors (strain and solutions), as well as the relevant post hoc analyses would be more relevant. They are however not presented.

In addition, classifying P rats as compulsive and Wistar as non-compulsive is an oversimplification of the data. Previous investigations have shown that compulsive-like behavior can be observed in Wistar animals, at least in some individuals (Augier et al., 2018; Hopf et al., 2010; Jadhav et al., 2017; Jadhav et al., 2018). Studying these behavioral processes at group level may be a clear limiting factor in understanding the molecular mechanisms mediating addiction-like processes. In agreement with this hypothesis, visual inspection of the data in Supplementary Figure 1 and Figure 1 seems to indicate that a substantial part of the P rats also stopped drinking alcohol after it was adulterated with quinine.

Compulsive vs non compulsive animals should therefore be defined using a more objective criterion, for example using a clear cut-off in the % difference in drinking compared to baseline, irrespective of the strain used or even better, cluster analysis ...

3) An important control experiment looking at quinine drinking alone is needed to rule out the possibility that differences between the groups could be explained by altered sensitivity to the taste of quinine alone in P rats compared to Wistars. Based on data in the literature showing higher drinking levels, self-administration and motivation for alcohol in P rats but also nicotine and non-drug rewards such as sweet solutions, it would not be unexpected that P rats could be less sensitive than Wistar to the bitter taste of quinine. It is also critical to show that the chemogenetic activation of dmPFC is not altering quinine drinking alone.

4) It is difficult for the reader to fully analyze and interpret the compulsive-like drinking data as, only ethanol intake in g/kg are shown. For a better comparison between the two groups, these data should also be represented as % decrease of alcohol intake compared to baseline. This is particularly important as P rats drink twice as much ethanol as Wistar at baseline.

5) Recent investigations have already started to characterize the role of the mPFC in aversive resistant-like alcohol drinking or self-administration, using both quinine-adulteration (Barbier et al., 2021) or footshock-punished self-administration (Halladay et al., 2020). These studies should be incorporated into the discussion of the present work.

6) The authors should test whether chemogenetic activation of dmPFC alters alcohol drinking per se and sucrose/saccharin quinine-adulterated drinking. This would strengthen the conclusion that activation of dmPFC prevents compulsive-alcohol drinking.

7) The manuscript would benefit of another round of editing to improve its clarity.

Minor comments:

8) The number of animals used for each experiment should be clearly indicated in the text and

figures

9) Did the authors also measure BECs during quinine adulterated drinking?

Augier, E, Barbier, E, Dulman, RS et al. (2018) A molecular mechanism for choosing alcohol over an alternative reward. *Science* 360: 1321-26.

Barbier, E, Barchiesi, R, Domi, A et al. (2021) Downregulation of Synaptotagmin 1 in the Prelimbic Cortex Drives Alcohol-Associated Behaviors in Rats. *Biol Psychiatry* 89: 398-406.

Halladay, LR, Kocharian, A, Piantadosi, PT et al. (2020) Prefrontal Regulation of Punished Ethanol Self-administration. *Biol Psychiatry* 87: 967-78.

Hopf, FW, Chang, SJ, Sparta, DR, Bowers, MS, Bonci, A (2010) Motivation for alcohol becomes resistant to quinine adulteration after 3 to 4 months of intermittent alcohol self-administration. *Alcohol Clin Exp Res* 34: 1565-73.

Jadhav, KS, Magistretti, PJ, Halfon, O, Augsburger, M, Boutrel, B (2017) A preclinical model for identifying rats at risk of alcohol use disorder. *Sci Rep* 7: 9454.

Jadhav, KS, Peterson, VL, Halfon, O et al. (2018) Gut microbiome correlates with altered striatal dopamine receptor expression in a model of compulsive alcohol seeking. *Neuropharmacology* 141: 249-59.

Dear Reviewers,

We would like to extend our sincere thanks to all the reviewers and the editor for your efforts reviewing our manuscript. We found the comments from the reviewers to be thorough, knowledgeable, and helpful. These are some of the most thorough and professional comments we have ever received on a manuscript. We agree with the vast majority of these comments and we have worked to improve the manuscript based on these suggestions. We believe your comments and the subsequent changes dramatically improved the paper. In the document below, we respond to each comment in turn (our responses are presented in **red text** and we have added comment numbers to help improve clarity). Thank you again for all your hard work!

Sincerely,
Nicholas Timme

Reviewer #1 (Remarks to the Author):

Timme et al, in their manuscript “Compulsive drinking is associated with neural activity patterns reflecting diminished behavioral control and enhanced seeking representations in dorsal medial prefrontal cortex”, examine how cortical populations differentially represent behaviors with respect to alcohol drinking. One purported phenotype of AUD is the continued consumption of alcohol in spite of associated negative consequences; and the hypothesis is that this is due to alterations in PFC circuits supporting decision-making. Understanding how these behavior-related computations have changes will shed light on alterations to mPFC contributions, and is an important question. Further, animals with different behavioral phenotypes are likely relying on different information to control behavioral output, so identifying which computations are different is important in order to assess their contribution to continued use.

Two strains of rats were exposed to initial alcohol consumption in their home cages, and the trained on a cued- alcohol availability task where CS+ predicted the soon-insertion of a alcohol sipper tube and CS- predicted absence. Animal tracking through deep lab cuts was used to nicely track animal position in the chamber. The authors were then able to use position in their analyses terming time spent by sipper outside CS+ and taking periods as “alcohol seeking”, akin to this reviewer as a modified conditioned place preference. Behaviors were quantified, and recording performed in mPFC populations of these two strains during 1) alcohol sessions and 2) alcohol + quinine sessions. They then showed that medial prefrontal cortex population activity correlates mapped onto principal component space showed different magnitudes of separation in relation to rat strain and challenged drinking. The majority of the manuscript is spent dissecting principal component separation in relation to a variety of behavioral variables, such as “seeking” behavior prior to the task cue. They show through primarily computational methodology how quinine drinking in non-compulsive animals is correlated with increased “engagement”, as quantified by principal component separation of mPFC population activity comparisons between trial types, in both seeking state and behavioral control representations. In contrast, quinine drinking in compulsive animals was correlated with greater “seeking representations”. Finally, they used a chemogenetic approach to prevent compulsive drinking observed in P-rats by increasing activity of mPFC to mimic the increased firing rate changes observed in non-compulsive animals during challenged drinking.

Overall, I am quite mixed, leaning more positive, on this manuscript. First, this is a beast of a paper, with a tremendous amount of interesting data. The authors should be commended for the computation approach taken; and indeed there is a lot of neural data presented in this paper (sometimes making it challenging to go through). The question asked is of interest, and the analyses showing the emergence of different representations (and how easily they were discriminable) were well-done. While I have concerns, they lay largely in more the interpretation and significance of the finding realm and I would like to hear what the authors response to the following is;

1) the determination of compulsivity is by strain, which raises the possibility that differences in representation differentiation may be a strain difference and not related to the “compulsive phenotype” as

This is a very good point that we should have discussed more clearly in the paper. In this work we determine compulsivity via behavior. However, the compulsive phenotype is closely related to the genotype, which is not unexpected given the known genotypic risk associated with AUD^{1,2}. Thus, we used strain as a proxy for compulsivity in almost all cases because the Wistars as a group demonstrated non-compulsive drinking and the P rats as a group demonstrated compulsive drinking. In the case of two later quinine tests, one Wistar recording met the criteria for being compulsive and one P rat recording met the criteria for being non-compulsive and these sessions were switched between groups (see Methods section (Track Changes Manuscript Lines: 483-485, 831-840, Clean Manuscript Lines: 437-439, 753-762) for added information about these criteria).

Furthermore, we believe it is unlikely that the differences we see in neural representations are driven solely by genotypic differences unrelated to the compulsive phenotype for several reasons. First, we detected differences in the neural firing behavior of a brain region known to be involved in decision-making and substance use disorders, and we detected these differences immediately before and during the expression of compulsive drinking, making it unlikely that these differences were merely genotypic artifacts unrelated to compulsive drinking. Furthermore, some of these differences were observed during epochs where no behavioral differences were observed between strains (e.g., approaching) or during epochs where the neural signal differences connected logically to behavior (e.g., large drink response in Wistars during quinine drinking which corresponded to a halt in drinking), indicating that these differences were not merely behavioral artifacts unrelated to compulsive drinking. In addition, there are several compelling reasons to use these strains for comparisons involving compulsive drinking. Recent work has shown P rats share gene expression differences with human AUD subjects³, which indicates that P rats are a highly relevant pre-clinical model for genetic risk for AUD. In addition, differences between P rats and Wistars are likely related to alcohol consumption, given that P rats were bred from Wistars. This common ancestry, along with the fact that Wistars can drink compulsively given sufficient drinking history⁴⁻⁸, implies that Wistars and P rats are very similar. That said, please see our response to your point 2 just below. We have added material to the introduction (Track Changes Manuscript Lines: 73-84, Clean Manuscript Lines: 70-81) and discussion (Track Changes Manuscript Lines: 655-673, Clean: 590-608) to better clarify this point.

2) very limited functional interrogation was performed to examine whether the emergence of these representations contributed to the ongoing behaviors (which would necessary to shown that seeking representations held within mPFC support compulsive seeking.

While we agree that further functional interrogation would be helpful, we believe that the chemogenetic interrogation that was performed provided crucial information that directly supported our behavioral and

neural population activity observations. This chemogenetic manipulation of dmPFC produced changes in compulsive drinking behavior in a model of genetic risk for AUD. Furthermore, this study identified key epochs of the decision-making process where neural activity differs in compulsive and non-compulsive subjects and it identified differences in neural activity (e.g., increased excitation in non-compulsive subjects). In this way, it went beyond functional studies that show dmPFC plays a role in compulsive drinking and instead identified neurocomputational differences between compulsive and non-compulsive subjects. Future experiments will leverage this information to design well-controlled functional manipulations targeted to these neurocomputational phenomena. Please also see our response below to a similar comment (Reviewer 1, Comment 5). That said, we agree that these points should have been better articulated in the manuscript, so we have added material to the discussion (Track Changes Manuscript Lines: 650-654, Clean Manuscript Lines: 584-589) to address this point.

3) While the PCA analyses are well-done and appreciated, I have some confusion given their selective presentation and the use of differentiation versus strength of contribution to behavior (no decoding). Point 2 may be beyond the scope of this already extensive body of work, but should be considered in the interpretation of what this is showing.

This is an excellent point. We agree that we should have discussed the distinction between our method and decoding/encoding analyses, and have therefore included one for this purpose using data structures that were identical to those used in the PCA. We added a paragraph near the end of the Methods Section explaining the differences between our PCA and a decoding/encoding analysis (Track Changes Manuscript Lines: 975-988, Clean Manuscript Lines: 897-910). This new paragraph also details the results of the decoding/encoding analysis and points to a new figure at the end of the Supplemental that presents the results of the analysis. We also included a sentence near Figure 2 to direct the reader to the decoding/encoding analysis (Track Changes Manuscript Lines: 210-211, Clean Manuscript Lines: 201-202). The decoding analyses further highlight the need to examine variables of interest at the population level as weak decoding was observed at the single neuron level but robust decoding was observed at the population level. Thank you for making this comment. We believe including this analysis significantly improves the paper.

4) Not clear is the justification for using PCA approach to examine activity differences between the two trial types as well as the two rat strains. While the authors address that they are not approximating strength of representations using trial classification methods, they do not make a strong case as to why they chose to use principal component analysis to quantify it instead. It is also unclear whether their main metric, “PC separation”, is indicative of a “strength” of representation, rather than a selectivity for those populations to respective trial types. In several figures, it is unclear whether their findings are contingent upon arbitrarily selected principal components, rather than those which consistently explain the majority of the variance in their data (e.g. top 3 principal components). While the authors make sure to demonstrate that 7 of the principal components are represented in a stable manner across their captured population, it is also unclear why they chose to include PC 10 in a later analysis. Finally, it is not fully clear what their PC analysis adds towards measuring “representation strength” that straightforward comparisons of peri-event population activity would not. The richness of mPFC population activity changes across their behavioral variables could be similarly, and more simply, shown through peri-event activity comparisons (as seen in Fig 2B-C, E).

We appreciate this comment and we agree that we should have more clearly articulated our rationale for using PCA. We chose to use PCA because we are interested in the dynamics of neural population activity throughout decision-making. Furthermore, there is compelling evidence that variables are better represented at the population level compared to the single neuron level in mPFC⁹, PCA is a common tool used to identify population signals within ensembles of neurons¹⁰, and the decoding analyses included in this revision further highlight the need to look at signals represented at the neural population level (Supplemental Figure 22). We have added text to the results section near Figure 2 (Track Changes Manuscript Lines: 198-200, Clean Manuscript Lines: 189-191) to better address our motivation for using PCA.

With regards to PC separation's relationship to the strength of the representation, it is possible that this is driven by differences in selectivity of various populations to certain trial types. However, it seems to us that this would be a natural means of increasing representation strength. In other words, if neurons are more selective in strain A vs. strain B, in our opinion, that would indicate increased representation strength in strain A. If we've misunderstood your comment, we apologize and we would appreciate it if you would clarify. We have added material to the results section near Figure 2 to clarify this point (Track Changes Manuscript Lines: 205-207, Clean Manuscript Lines: 196-198).

With regards to the point about different figures using different PCs, we apologize that this was not clear. We utilized the same PCs for all analyses. We show 3 example PCs with large separation effect sizes in Figures 3-6 to illustrate how PC separations relate to the representations assessed in each figure. These PCs were selected based on the magnitude of their separation during the epochs of interest. Furthermore, PC 10 was found to be stable, so it was utilized in the analysis (see Figure 2f). When we compared between strains, we utilize all seven stable PCs (PCs 1-6 and 10). We added material to the figure captions for Figures 3-6 to clarify this point.

With regards to peri-event activity comparisons, given the heterogeneity of neural firing patterns during decision-making in dmPFC, peri-event activity traces of individual neurons are difficult to categorize. Furthermore, while showing data for individual neurons can provide helpful, qualitative evidence that the statistical approaches used in the analysis capture the firing properties of the neurons measured, doing so can also lead to misconceptions about how robust the firing response is and the kind of information encoded in these responses. To this point, we show peri-event activity graphs in Figure 2e and 3f to demonstrate that a wide variety of response patterns are evident in the neural firing patterns when they are sorted by contributions to different PCs. To more directly address this point in the text, we added material to the results section near Figure 2 (Track Changes Manuscript Lines: 223-227, Clean Manuscript Lines: 205-209). Furthermore, the newly added encoding/decoding experiment demonstrates that a small portion of individual neurons do represent some of the decision-making relevant signals of interest in this analysis (see Supplemental Figure 22).

5) It is hard to determine how much of the findings is due to strain comparison versus the emergence of a compulsive phenotype of mPFC representations. Classification of compulsivity is based on genotype (P rats bred to excessively drink alcohol)- with the compulsive behavior suggested as an endophenotype. Compulsive drinking is measured as the continued consumption of quinine-adulterated alcohol. Wistar rats (drink less alcohol at baseline) were used for comparison as a group who is more sensitive to quinine adulteration. So, in essence this is a strain comparison, and consumption led to blood ethanol concentrations that ranged between quite low, to slightly above the legal limit. Not clear to this reviewer

is whether representation of task information is different between the two independent of a) the prior alcohol consumption history and b) use of alcohol as the reinforcer. Is that the different genotypes use different types of information during performance of the task (i.e. strain effect)? Is the argument that that emergence of the compulsive phenotype is going to coincide with emergence of these patterns of representations? There were no specific perturbations to mPFC activity during computation-specific epochs to assess function of these representational patterns to the support of the ongoing behavior. Does inhibition of mPFC activity in Wistar rats increase the consumption of quinine? Why use the two strains in this way?

Thank you for these comments! With regards to the points about genotype vs. phenotype, please see our response to your previous comment about this issue (Reviewer 1, Comment 1).

With regards to the question about prior alcohol consumption history, this is a good point. While P rats and Wistars were nearly matched for number of drinking sessions (for first quinine exposure: 6 IAP sessions, 10 2CAP training sessions, ~4-6 2CAP ephys sessions), consumption values differed between the strains (see Supplemental Figure 16). It is important to note that other models of compulsive alcohol consumption in rats (e.g., long drinking history Wistars) would involve greater differences in lifetime alcohol consumption and number of drinking sessions between the treatment group and the control group. We have added material to the discussion section to address this point (Track Changes Manuscript Lines: 686-706, Clean Manuscript Lines: 621-630) and we have better clarified Supplemental Figure 16.

With regards to the impact of using alcohol as a reinforcer, we believe it would be interesting to examine this question further in future studies by including other reinforcers, but we feel it is beyond the scope of this work and outside our specific topic of interest: compulsive alcohol consumption.

With regards to the point about specific perturbations of dmPFC during computation-specific epochs, we believe it is important to emphasize that this study identified those computation-specific epochs. Also, please see our response to a similar comment above (Reviewer 1, Comment 2). We have added material near the end of the discussion (Track Changes Manuscript Lines: 650-654, Clean Manuscript Lines: 584-589) to better address this point.

With regards to the point about inhibiting Wistar dmPFC in an attempt to increase compulsive drinking, we agree that this could be a helpful experiment. However, there are several subtleties that must be considered when designing this experiment. To test the hypotheses generated herein, real time feedback procedures would need to be developed to drive optogenetic stimulation. For example, if these experiments are to be done properly, one would need to be able to detect when an animal is approaching the sipper and then inactivate in real time. If one was to broadly inactivate during the entire drinking epoch off-target processes would likely be inhibited and therefore complicating the interpretation of the results. However, we agree that this type of experiment is critical and while developing these approaches in our group have been slowed by the pandemic, we are working towards this goal. Further, we have added material to the discussion to highlight the importance of this critical next step (Track Changes Manuscript Lines: 652-654, Clean Manuscript Lines: 587-589).

With regards to your general point about why we would use these strains, we were generally influenced by studies of human subjects that compare family history positive and family history negative subjects as a means to uncover innate neurobiological differences as a function of genetic risk (e.g., ^{11, 12}). Given the relevance of P rats as a pre-clinical model for genetic risk for AUD (e.g., ³), we felt Wistars were an

excellent comparison strain given their common ancestry (P rats were derived from Wistars¹³) and Wistars can drink compulsively given a sufficiently long drinking history⁴⁻⁸. We have added material to the introduction to better communicate our motivation and to the discussion (Track Changes Manuscript Lines: 655-673, Clean Manuscript Lines: 590-608) addressing the strength of the P rat/Wistar comparison.

6) In relation to the above; all initial analyses were based on genotype- and not defined by a particular phenotype; however, two subjects were switched between groups based on phenotype? This is problematic, and one criteria should be used for group inclusion.

Yes, two recordings were switched between groups based on phenotype for later recordings. To clarify this issue, please see our responses above regarding genotype vs. phenotype (Reviewer 1, Comment 1).

7) Statistics do not always support the findings. For example, in Fig. 1e, examining consumption of quinine alcohol solution, there is a main effect of strain, and a main effect of quinine, but no interaction. So- overall, P rats drank more than Wistars, and overall, there was less quinine solution consumed, but not a difference in patterns of consumption suggested, “P rats showed a smaller reduction in consumption” is not supported. The claims or statements made in much of paragraph 125-142 are not supported statistically.

We agree that these analyses should be improved. Therefore, we added a comparison of percent decrease in consumption to show that P rats had a smaller reduction in consumption (new Fig. 1f). We also clarified the role of the post-hoc test in the figure caption for Fig. 1e. We note that we are entitled to probe for these effects via post hoc test as 1) we have a strong a prior hypothesis (it replicates a prior finding in our lab¹⁴) and 2) more importantly, each of the factors of the omnibus ANOVA are significant. Regarding the quinine comparison test in Supplemental Figure 1, we clarified the figure legend and added a statistical analysis to demonstrate quinine consumption decreased. Regarding relative approaches between CS+ and CS- trials, we added a statistical analysis to demonstrate that the CS sensitivity was greater than 0.5 in all but one group (Fig. 1j). Regarding sipper occupancy through time (Fig. 1k), we added an analysis to demonstrate that Wistars on quinine days were significantly less likely to occupy the sipper during later access times. These changes can be found in Figure 1, the Figure 1 caption, and in the main text (Track Changes Manuscript Line: 164, Clean Manuscript Line: 159).

8) Further, for the DREADD experiments, the supporting statistics are very unclear. The first test should similar sensitivity to quinine and maybe a reduction from baseline (all determined visually)- then control tests were done showing variability in the quinine adulteration effect. Then, more CNO and quinine tests were performed. What is reported statistically for all of this is a there is a main effect of DREADD group; there seems to be many things going on in this within subject design. One is the separation of statistical analyses for the first test should be justified; second, the differences in the control tests need to be examined, and lastly, the continued test analysis does not support a pattern difference but an overall main effect of CNO. However, all of the above are just reporting group differences, with no analyses performed against baseline (1.0) (do they actually reduce drinking). There are group differences as well as magnitude effects on sensitivity to quinine (which the P rats seem to reduce their drinking by 50 %- not an insignificant amount and questions the assignment of the compulsivity label).

We agree that the statistics for the DREADD experiments should have been more clear. Therefore, we reorganized the relevant paragraph of the Results Section and added several statistical analyses to address

the points you made. We added direct comparisons between DREADD and control animals on the first CNO+Quinine test, as well as change in drinking from baseline. We clarified the main effect of DREADD over all the CNO+quinine tests. For all three control tests, we added direct comparisons to the first CNO+quinine test and change in drinking from baseline. These changes can be found in Figure 7, the Figure 7 caption, and in the main text (Track Changes Manuscript Lines: 509-538, Clean Manuscript Lines: 462-484).

9) More of the data used to support the approach to sipper as seeking should be reported. That animals rarely stayed at one sipper between trials or failed to drink during access has no data associated with it. Please provide data supporting these claims.

This is a very good point. Regarding the point of staying at a sipper between trials, we added a reference to the relevant portion of the results section to the occupancy graph in Figure 1. This shows a low likelihood to be occupying the sipper before the trial starts. Regarding the point of failing to drink during access, we better quantified this behavior and found that 95.08% of correct approach trials resulted in a least one lick. We clarified this text and moved it to the Figure 3 caption. We also added material to the methods section (Track Changes Manuscript Lines: 930-931, Clean Manuscript Lines: 852-853) to address this point.

10) Change point analyses: Interesting approach, more data is needed in the average number of change points exhibited- or the density of change point occurrences across the session. Did the authors just go with the first one? Looking at example Fig. 3c, it seems rats could have multiple change points? Given the alignment of neural activity, these details would be informative.

These are excellent questions. We used the largest magnitude change point, per the MATLAB function `findchangepts.m`. We updated the relevant portions of the results section (Track Changes Manuscript Lines: 273, 385, 422, 896-897, 917, Clean Manuscript Lines: 249, 351, 378, 818-819, 839) and the methods to make this point clearer.

11) Wave-form identification of cortical neuron type can be problematic; please use putative; not sure that such identification is necessary

Yes, we agree with this comment. We should have qualified these neurons as putative. We have made this change to throughout the manuscript and we added an explanation of this qualification (Track Changes Manuscript Lines: 488-490, Clean Manuscript Lines: 442-444). We still feel this analysis adds to the paper by pointing to a possible roll for inhibitory and excitatory neurons in compulsive drinking, so we have left it in place.

12) Is there concern that the use of DMSO, that produces long lasting smell/perhaps taste properties interfered with sensory (smell/taste) aspects contributing to alcohol consumption?

This is a valid concern, but our control CNO+alcohol test (which showed a small decrease in drinking relative to baseline and a smaller decrease in drinking in comparison to CNO+quinine or Vehicle+quinine tests) demonstrates that the smell of DMSO does not have a large effect on alcohol consumption in these circumstances.

13) 53 - can authors clarify what role they are referring to regarding mPFC and compulsive drinking?

That paper did not assess compulsive drinking, so we have clarified the sentence to make this point clearer (Track Changes Manuscript Lines: 56-57, Clean Manuscript Lines: 54-55).

14) 67 - clarify what is meant by "some drinking sessions" in relation to experimental design

We have reworded that sentence to make it more clear (Track Changes Manuscript Line: 72, Clean Manuscript Line: 69).

15) could benefit from short discussion of what PCA approach could offer "regarding large-scale patterns of neural activity evolve during the compulsive decision to drink" that perievent activity does not.

Please see our response to your earlier comment about peri-event activity analyses (Reviewer 1, Comments 4).

16) 149-152 - clarify epochs were defined by observer (e.g camera)

While it is true that some epochs were defined behaviorally via analysis of the tracking data (e.g., sipper approach and drinking), the other epochs were defined by the task itself. For instance, the cue epoch was defined by the time when the cue was presented. We have added material near that portion of the paper (Track Changes Manuscript Lines: 193-195, Clean Manuscript Lines: 185-186) to make this point clearer.

17) 159-161 "Note that we do not attempt to identify which trial represents the signal, rather we only quantify separation as a measure of representation strength." what is the argument for not quantifying representation strength via trial identification (e.g. decoding analysis)?

Please see our response above to your first comment about decoding (Reviewer 1, Comment 3).

18) Fig 2b/c add y-scale or something, what do reductions mean?

The y-scale represents baseline subtracted and scaled firing rate, so reductions in the y-scale are reductions in firing rate. We have added a bar to communicate the y-scale more clearly and we have added material to the figure caption.

19) While the authors control for neuron yield across experimental groups, it would be of interest to see if differences observed can be explained by animal preference/aversion to the stimuli, e.g. do a couple of animals whose neurons have strong selectivity for alcohol/quinine drive the observed distances differences between population trial type PCs?

We agree that it is important to assess the degree to which an individual will drive certain results in this type of analysis. Therefore, we conducted a leave-one-out analysis (Track Changes Manuscript Lines: 969-974, Clean Manuscript Lines: 891-896) and found that the results were largely identical. More specifically, of the 14 animals in this study, removing 12 (one a time) resulted in no changes to the number of stable PCs or statistical changes to the PC separation comparisons (15 comparisons were performed for the first quinine test sequence). (Here, we define "statistical changes" as those statistical comparisons that switch from significant to not significant, vice versa, or switch directions of differences (e.g., PQ decreases instead of increases).) For one P rat, leaving it out resulted in one fewer stable PC and

3 statistical changes to PC separation comparisons. For one Wistar, leaving it out resulted in two additional stable PCs and 3 statistical changes to PC separation comparisons. Based on the results of this leave-one-out analysis and the few differences it produced, we feel it would not be possible to gain further insights into the data by breaking this analysis out into individual animals.

20) Fig 3E unclear what the randomized data is on panel

We have added material to the figure caption to address this point.

21) 253 attempt at explaining how variance explained by data in corresponding PC space is indicative of "strength" of a representation (but why not just compare firing rates of that neuron to the to trial types? what does PCA add?)

We apologize, but we do not understand your comment. Perhaps 253 is the incorrect line number? On line 253 we are attempting to clarify that representation strength is not necessarily the same thing as the strength of the subjective wanting experienced by the animal. Please see our comments above regarding our motivation for using PCA (Reviewer 1, Comment 3).

22) Note: comparisons of PC separation within epochs could be confirmed further by using permutation tests between the two traces (e.g. Fig 3H,I), may even serve to show exactly when in these windows these differences exist

Thank you for this suggestion. However, we don't believe this would be appropriate due to the nature of the trials that would be permuted in such an analysis. Figure 3H (for instance) plots the mean and std of PC separations across subsampling trials. The number of subsampling trials is large (500), so we suspect that most PC separations would be found to be significantly different between groups, regardless of effect size, if we permuted the subsample trials between groups due to the large power supplied by the number of subsampling trials. Instead, we feel our current method is more appropriate. We compare the mean PC separation values across the time bins in the epoch of interest (20 time bins). In this way, the power supplied by the large number of subsampling trials increases confidence in the mean PC separation value in each time bin and comparisons between groups assess the stability of the differences in PC separation throughout the epoch.

23) Note: Confused as to why PC comparisons were made between different PCs (e.g. PC 1vs2vs4 in Fig 3, PC 1vs3vs5 in Fig 4 etc), do PC separation analysis rely on these hand-picked PC comparisons? Why not keep analysis consistent to, say, the top 3 PCs? It's confusing because if I compare PC 1 (e.g. 10% variance explained) to PC 6 (e.g. 1% variance explained) then it is not surprising to see these difference, hence why justification for PCs shown/analysis may be required. In the text, or figure legend, it is unclear if PC separation is being made for the PCs show in the 3D trajectory or for all 7 shown to be stable.

We have added material to the captions for Figures 3-6 to address this point. Please see our response above to a similar question for more details (Reviewer 1, Comment 4).

24) Fig 5 Why bring PC 10 into analysis if only the first 7 were identified as stable?

Only 7 PCs were stable, but not the first 7 (see Figure 2f). We have added material to the caption for Figure 2 to make this clearer.

25) Figure 5 E-G Y axis meant to show PC score? if so, label appropriately and include in legend, not clear what is on the y scale

Yes, it is the PC score, thank you for pointing this out. We have added material to the figure and the caption to clarify this point.

Reviewer #2 (Remarks to the Author):

In this study, Timme and colleagues used novel principal components and change point analyses to evaluate neural population representations in the dorsomedial prefrontal cortex (dmPFC) associated with compulsive alcohol drinking in rats. Results revealed several key differences in neural representations for behavioral control and alcohol consumption that could explain differences between compulsive and non-compulsive drinking rats. For instance, neural representations for alcohol seeking states were enhanced in compulsive drinkers. On the other hand, neural representations for cues associated to alcohol availability, as well as for approach initiation to the alcohol sipper and during actual drinking periods were diminished in the compulsive drinkers. In addition to these findings on neural representations, Timme and colleagues tested the idea that changes in neural activity in the dmPFC using chemogenetic-mediated manipulations could alter escalation of alcohol drinking in the compulsive rats. They found that chemogenetic-based excitation of dmPFC activity prevented escalation of compulsive drinking. Collectively, these findings support that notion that differences in neural activity profiles in the dmPFC underlie compulsive alcohol seeking and drinking.

Overall, I feel enthusiastic with the prospect of publication for this article. The text is very well written and organized, experiments are well controlled, and the data and results are very informative and well illustrated. Though, in the list below I highlight a few issues that require revision for improvement of the article and to further strengthen the main conclusions.

1. In Fig 7 (panels c and f), analyses on spike waveforms were performed to differentiate between excitatory versus inhibitory neurons. While this method used to be quite acceptable in the past, more recent evidence suggests that spike widths are not necessarily an accurate method to tell excitatory versus inhibitory cells apart. For instance, while PV+ interneurons could have significantly shorter spike widths, SOM+ interneurons could have wider spike widths similar to principal neurons (Wolff et al., 2014; <https://pubmed.ncbi.nlm.nih.gov/24814341/>). In addition, some subpopulations of CCK+ interneurons also show wide spike widths (Vogel et al., 2016; <https://pubmed.ncbi.nlm.nih.gov/27497223/>). Thus, to avoid confusion or potentially misleading interpretations, I suggest moving this analysis on “excitatory” versus “inhibitory” neurons to supplemental materials. In addition, the authors should consider using the terms “putative excitatory” versus “putative inhibitory” neurons.

Thank you for sharing these references and we agree that this information should be included in the paper. As suggested, we have switched to referring to these as putative excitatory/inhibitory neurons, we have included citations to these papers, and we describe why we refer to these as putative excitatory/inhibitory neurons (Track Changes Manuscript Lines: 488-490, Clean Manuscript Lines: 442-444). We feel this

analysis still provides very useful information to the reader, though, so we have decided to leave it in the main text.

2. The rationale for the chemogenetic experiment in Fig 7g-i (lines 418-421) makes much sense in terms of the findings obtained from electrophysiology. In addition, the authors obtained findings that are consistent with the rationale (i.e., chemogenetic-mediated excitation of dmPFC prevented escalation of compulsive drinking in the P-rats). Though, the main conclusions of this article would get strengthened even further, perhaps to an almost irrefutable point, if the authors also perform the opposite experiment, i.e., chemogenetic-mediated inhibition of dmPFC in the Wistar rats to artificially induce compulsive drinking.

We certainly agree that using a chemogenetic approach to inhibit dmPFC in Wistars to artificially induce compulsive drinking would be a very valuable experiment. However, there is a potential issue that gives us pause with this experiment. The ability to restore a pathological behavior is often more valuable from a treatment perspective than generating a pathological state – thus our focus on “treating” the P rats. When generating a model of a pathological state (e.g. generating a compulsive drinking animal) it is critical that the model retain “computational validity”, where the computations that lead to the altered behavioral phenotype are implemented in a model system as precisely as possible. In our work, we have identified several computations in dmPFC that characterize the compulsive phenotype. Our critical next step is to precisely implement these altered computations in Wistars to drive compulsive drinking. Broad inhibition of the dmPFC via DREADD, may result in compulsive phenotype, but without simultaneous neural recordings this would not provide insight into the impaired computations leading to compulsive drinking. Unfortunately, due to the pandemic, this experiment has been delayed and we feel it is appropriate to move forward without it. We definitely agree that this is an important point, so we have added material to the discussion section to address this point (Track Changes Manuscript Lines: 650-654, Clean Manuscript Lines: 584-589).

3. In Supp-Fig 19, it can be appreciated that some electrode placement tracks reached deeper areas considered as ventral mPFC. Where neurons from these excluded?

Excellent question. It is important to note that the tracks represent the final position of the probes, which were mounted on movable drives. Based on how frequently the probes were moved and the distance traveled, the initial quinine test recordings (which form the bulk of the analysis) occurred ~300-500 um above the final positions shown in the tracks. The neurons from these more ventral probes were not excluded from the analysis. We have added material to caption of for Supplemental Figure 19 to address this point.

Anthony Burgos-Robles, PhD
University of Texas at San Antonio

Reviewer #3 (Remarks to the Author):

The report by Timme and colleagues describes the role of the dorsal medial prefrontal cortex (dmPFC) in controlling aversive-resistant drinking in a strain of rats selectively bred for high-alcohol preference. They use an interesting PCA-based method to study neural representation and report that chemogenetic activation of the dmPFC prevented the emergence of compulsive-like drinking. While the study is

interesting, there are some major concerns that reduce enthusiasm for the manuscript. The main finding indicating a role of dmPFC in aversive-resistant drinking is also not entirely novel.

Thank you very much for taking the time to review our manuscript and for supplying these valuable comments! Regarding your overall synopsis of the main finding of the manuscript, we agree that we are not the first group to implicate a critical role of the dmPFC in aversion-resistant drinking. Studies outlining a role for the mPFC provide the rationale for examining how the computation is altered in this brain region. Our contribution and main findings identify these altered computations in dmPFC.

The validity of the compulsivity model presented by the authors is a major concern:

1) The numbers of animals used for the main behavioral experiment (Figure 1) is very low (not clearly indicated in the manuscript, but visual inspection of the figure seems to indicate a $n=6$ or 7 per group), ultimately leading to a low power for this analysis. This experiment requires more subjects to strengthen the conclusions.

We agree that we should have been more clear that we only gathered and were only showing compulsive drinking behavior data for the electrophysiology animals. The number of electrophysiology animals (7 Wistars and 7 P rats) in this study lead to a very large number of neurons for analysis and, furthermore these sample sizes are well within accepted range for this type of experiment¹⁵⁻¹⁸. In addition, there is a strong a priori hypothesis for the behavior as these behavioral data replicate previous behavioral studies performed in our lab that showed similar compulsive and non-compulsive drinking behaviors from these strains of rat¹⁴. We have added material to the Results section near Figure 1 to clarify the number of animals and their role in the experiment (Track Changes Manuscript Lines: 128-131, Clean Manuscript Lines: 125-128). Furthermore, we better clarified in the introduction that previous behavioral studies have demonstrated compulsive drinking in P rats and non-compulsive drinking in Wistars in this type of task (Track Changes Manuscript Lines: 73-75, Clean Manuscript Lines: 70-72).

Also, only one session was performed to assess baseline drinking levels, as well as the compulsive-like drinking using quinine adulteration. This is insufficient to confirm that a stable behavior was established and can be trusted to guide the following molecular investigations performed by the authors. Even more for the quinine adulteration experiment, drinking during the first day of exposure is very sensitive to the immediate aversive taste of quinine and can differ greatly from the stabilized drinking levels after at least a few sessions have been performed, in particular for Wistar rats.

We agree that the stability of the behavior is an important concern, but we also believe it is important to consider acclimation effects that may bias results. Our goal in focusing on the first day of quinine was to avoid the development of tolerance to quinine's aversive properties in either strain for the primary analysis. It should be noted that later quinine tests are shown and analyzed in the manuscript (see Figure 7) and that compulsive drinking was relatively stable in these later sessions (see Supplemental Figure 12). We have added material to the manuscript near Figure 1 to clarify the use of additional compulsive drinking tests and our motivation for focusing on the first quinine exposure (Track Changes Manuscript Lines: 123-128, Clean Manuscript Lines: 119-125).

2) The definition of the compulsive and non-compulsive groups is also an important issue. There are no clear criteria on how the authors classify compulsive vs non-compulsive animals, except the fact that there was a main effect of strain and solutions in Figure 1. The interaction between the two factors (strain

and solutions), as well as the relevant post hoc analyses would be more relevant. They are however not presented.

We apologize for the lack of clarity on these issues. Regarding the criteria for how we define and classify compulsive vs. non-compulsive animals, we have added material near Figure 1 (Track Changes Manuscript Lines: 170-173, Clean Manuscript Lines: 164-167) to clarify that we define the phenotype behaviorally, but that there was a substantial alignment between the phenotypic split and the strain. We also added material to the methods section to better describe the criteria for compulsive/non-compulsive classification (Track Changes Manuscript Lines: 832-840, Clean Manuscript Lines: 755-762).

Regarding the comments about the statistical tests, we agree that this should be more clear. In response to this comment and a comment from another reviewer (Reviewer 1, Comment 7), we added additional analyses and explanation to Figure 1. Since we have previously observed behavioral differences in quinine consumption between these lines, we performed an analyses of the difference between solutions within Wistars and found that it was significant, while it was not in P rats. This result is shown in the figure, but it was not clearly discussed in the figure caption, so we better clarified this result. We added a new figure (Fig. 1f) and accompanying statistical tests to directly compare percent change in consumption. We also added statistical tests to Fig. 1k to compare sipper occupancy behavior to demonstrate that Wistars left the sipper during quinine sessions and P rats did not, thus indicating behavioral differences in seeking.

In addition, classifying P rats as compulsive and Wistar as non-compulsive is an over-simplification of the data. Previous investigations have shown that compulsive-like behavior can be observed in Wistar animals, at least in some individuals (Augier et al., 2018; Hopf et al., 2010; Jadhav et al., 2017; Jadhav et al., 2018). Studying these behavioral processes at group level may be a clear limiting factor in understanding the molecular mechanisms mediating addiction-like processes. In agreement with this hypothesis, visual inspection of the data in Supplementary Figure 1 and Figure 1 seems to indicate that a substantial part of the P rats also stopped drinking alcohol after it was adulterated with quinine. Compulsive vs non compulsive animals should therefore be defined using a more objective criterion, for example using a clear cut-off in the % difference in drinking compared to baseline, irrespective of the strain used or even better, cluster analysis ...

Thank you for these insightful comments! With regards to compulsive-like behavior in Wistars, we agree that this has been observed. However, as far as we are aware, those studies all involve substantially longer drinking histories (e.g., 3-4 months⁴) than were used for the Wistars in our study. We have added material, including references to those works that we had not previously referenced, to the Introduction (Track Changes Manuscript Lines: 73-84, Clean Manuscript Lines: 70-81) and Discussion (Track Changes Manuscript Lines: 686-706, Clean Manuscript Lines: 621-630) to more thoroughly address long drinking history Wistars.

With regards to studying processes at the group level, we agree that it would be ideal to incorporate more individualized data into the analysis. However, doing so is difficult when also attempting to analyze neural population data given the effects of neuron yield in such analyses. We did utilize individual behavioral data in determining compulsive/non-compulsive groups (e.g., two recordings were switched between groups).

With regards to decreases in consumption in P rats when alcohol was adulterated with quinine, we agree that this should have been clearer. In Figure 1, we show that P rat drinking appears to decrease (though not significantly). We have also added an analysis (new Figure 1f) that shows Wistars had a larger decrease in consumption with quinine. Furthermore, in Supplemental Figure 1, we show that P rat consumption of quinine adulterated alcohol decreases when given the option to drink unadulterated alcohol. This demonstrates that P rats were capable of tasting the quinine and found it aversive. We have improved Supplemental Figure 1 to make this point clearer. Overall, we believe these data demonstrate an important behavioral distinction between strains that justifies the use of strain to identify the compulsive phenotype, especially during the first exposure.

With regards setting a more objective criterion, we agree with you. Therefore, we used thresholds in the z-scored alcohol+quinine consumption and percent decrease in consumption during alcohol+quinine sessions to identify a compulsive Wistar session and a non-compulsive P rat session. Thus, it is important to note that we did not rely only on strain to determine compulsive/non-compulsive groups. We have added material to the Supplemental Figure 14 and the Methods section (Track Changes Manuscript Lines: 832-840, Clean Manuscript Lines: 755-762) to make this point clearer.

3) An important control experiment looking at quinine drinking alone is needed to rule out the possibility that differences between the groups could be explained by altered sensitivity to the taste of quinine alone in P rats compared to Wistars. Based on data in the literature showing higher drinking levels, self-administration and motivation for alcohol in P rats but also nicotine and non-drug rewards such as sweet solutions, it would not be unexpected that P rats could be less sensitive than Wistar to the bitter taste of quinine. It is also critical to show that the chemogenetic activation of dmPFC is not altering quinine drinking alone.

This is a good point that we should have better addressed in the paper. We had the same initial concerns and performed a preference test that demonstrated that P rats prefer alcohol to alcohol+quinine (see Supplemental Figure 1). So, P rats were able to taste the quinine and found it aversive. We have improved Supplemental Figure 1 to make this point clearer. Furthermore, as we discuss in the manuscript near Figure 1, past research has demonstrated that P rats exhibit elevated aversive facial responses in response to tasting quinine¹⁹. We did not perform experiments on these animals to assess quinine consumption outside the presence of alcohol. We are unsure if testing quinine sensitivity in water (for instance) is necessarily relevant to quinine sensitivity in alcohol given synergistic effects of the alcohol+quinine taste as well as logistical hurdles to interpretation of such results. Such experiments would produce a raft of other questions about the context in which the water+quinine is consumed (e.g., in the task or in home cage) and how to motivate the animal to consume water alone (e.g., use of water restriction). All of these questions could produce substantial confounds. Furthermore, if it were true that P rats found quinine aversive, but not as aversive as Wistars and all differences we observe were due to this difference in early taste sensory systems, we would expect neural representation results for P rats drinking quinine to show similar, but weaker, changes relative to P rats drinking alcohol as were observed between Wistars drinking alcohol and alcohol+quinine. This was not what we observed. Often, the addition of quinine altered neural representations in P rats and Wistars independently of the patterns that were observed for alcohol alone. That said, we agree that it is possible the compulsive drinking results in P rats are influenced by a different taste sensitivity to quinine. Therefore, we have added material to the discussion section to directly address this point (Track Changes Manuscript Lines: 674-685, Clean Manuscript Lines: 609-620).

Regarding chemogenetic activation of dmPFC altering quinine drinking alone (i.e., in water), we feel this would present similar logistical and interpretation concerns we discussed above. Furthermore, the existing DREADD data demonstrate that there is no acute effect of chemogenetic activation of dmPFC on alcohol+quinine drinking (CNO+Qui Test 1, Figure 7h) and, therefore, presumably it has no acute effect on quinine sensitivity. Furthermore, comparing CNO+Qui Test 1 to the control test Veh+Qui for control animals shows that CNO does not alter alcohol+quinine drinking and, therefore, presumably does not acutely affect quinine sensitivity.

4) It is difficult for the reader to fully analyze and interpret the compulsive-like drinking data as, only ethanol intake in g/kg are shown. For a better comparison between the two groups, these data should also be represented as % decrease of alcohol intake compared to baseline. This is particularly important as P rats drink twice as much ethanol as Wistar at baseline.

We agree. We have added the relevant plot as Figure 1f.

5) Recent investigations have already started to characterize the role of the mPFC in aversive resistant-like alcohol drinking or self-administration, using both quinine-adulteration (Barbier et al., 2021) or footshock-punished self-administration (Halladay et al., 2020). These studies should be incorporated into the discussion of the present work.

Thank you for sharing these works with us! The Barbier et. al. is particularly relevant. We have added these papers to the manuscript (Track Changes Manuscript Lines: 44, 647-650, Clean Manuscript Lines: 42, 581-584).

6) The authors should test whether chemogenetic activation of dmPFC alters alcohol drinking per se and sucrose/saccharin quinine-adulterated drinking. This would strengthen the conclusion that activation of dmPFC prevents compulsive-alcohol drinking.

Regarding chemogenetic activation of dmPFC and alcohol drinking, we performed that control experiment and it is shown in Figure 7I (drinking was only slightly affected). We did not assess sucrose/saccharin drinking in any form with these animals as our primary focus is compulsive alcohol consumption.

7) The manuscript would benefit of another round of editing to improve its clarity.

We have further edited the manuscript for clarity. In addition, the comments provided by you and the other reviewers have greatly helped to improve the manuscript. Thank you!

Minor comments:

8) The number of animals used for each experiment should be clearly indicated in the text and figures

We added a sentence near figure 1 (Track Changes Manuscript Lines: 128-131, Clean Manuscript Lines: 125-128) that clearly states how many animals were used in the main experiment and pointing the reader the Methods section where the number of animals is discussed in detail. We added number of animal information to figures where appropriate.

9) Did the authors also measure BECs during quinine adulterated drinking?

No, we did not collect blood samples for these experiments out of concern that they might impact behavior on subsequent alcohol+quinine drinking sessions. We have clarified this in the methods section where we discuss the BEC measurements (Track Changes Manuscript Lines: 771-773, Clean Manuscript Lines: 694-697).

Augier, E, Barbier, E, Dulman, RS et al. (2018) A molecular mechanism for choosing alcohol over an alternative reward. *Science* 360: 1321-26.

Barbier, E, Barchiesi, R, Domi, A et al. (2021) Downregulation of Synaptotagmin 1 in the Prelimbic Cortex Drives Alcohol-Associated Behaviors in Rats. *Biol Psychiatry* 89: 398-406.

Halladay, LR, Kocharian, A, Piantadosi, PT et al. (2020) Prefrontal Regulation of Punished Ethanol Self-administration. *Biol Psychiatry* 87: 967-78.

Hopf, FW, Chang, SJ, Sparta, DR, Bowers, MS, Bonci, A (2010) Motivation for alcohol becomes resistant to quinine adulteration after 3 to 4 months of intermittent alcohol self-administration. *Alcohol Clin Exp Res* 34: 1565-73.

Jadhav, KS, Magistretti, PJ, Halfon, O, Augsburger, M, Boutrel, B (2017) A preclinical model for identifying rats at risk of alcohol use disorder. *Sci Rep* 7: 9454.

Jadhav, KS, Peterson, VL, Halfon, O et al. (2018) Gut microbiome correlates with altered striatal dopamine receptor expression in a model of compulsive alcohol seeking. *Neuropharmacology* 141: 249-59.

Citations

1. Moss, H.B., Chen, C.M. & Yi, H. Subtypes of alcohol dependence in a nationally representative sample. *Drug and Alcohol Dependence* **91**, 149-158 (2007).
2. Nurnberger, J.I., et al. A Family Study of Alcohol Dependence: Coaggregation of Multiple Disorders in Relatives of Alcohol-Dependent Proband. *Archives of General Psychiatry* **61**, 1246-1256 (2004).
3. Walker, L.C., et al. Acetylcholine Muscarinic M4 Receptors as a Therapeutic Target for Alcohol Use Disorder: Converging Evidence From Humans and Rodents. *Biological Psychiatry* **88**, 898-909 (2020).
4. Hopf, F.W., Chang, S.J., Sparta, D.R., Bowers, M.S. & Bonci, A. Motivation for alcohol becomes resistant to quinine adulteration after 3 to 4 months of intermittent alcohol self-administration. *Alcoholism, Clinical and Experimental Research* **34**, 1565-1573 (2010).
5. Hopf, F.W. & Lesscher, H.M.B. Rodent models for compulsive alcohol intake. *Alcohol* **48**, 253-264 (2014).
6. Augier, E., et al. A molecular mechanism for choosing alcohol over an alternative reward. *Science* **360**, 1321-1326 (2018).
7. Jadhav, K.S., Magistretti, P.J., Halfon, O., Augsburger, M. & Boutrel, B. A preclinical model for identifying rats at risk of alcohol use disorder. *Scientific Reports* **7**, 9454 (2017).
8. Jadhav, K.S., et al. Gut microbiome correlates with altered striatal dopamine receptor expression in a model of compulsive alcohol seeking. *Neuropharmacology* **141**, 249-259 (2018).
9. Durtsewitz, D., Vittoz, N.M., Floresco, S.B. & Seamans, J.K. Abrupt transitions between prefrontal neural ensemble states accompany behavioral transitions during rule learning. *Neuron* **66**, 438-448 (2010).

10. Cunningham, J.P. & Yu, B.M. Dimensionality reduction for large-scale neural recordings. *Nature Neuroscience* **17**, 1500-1509 (2014).
11. Dager, A.D., *et al.* Influence of alcohol use and family history of alcoholism on neural response to alcohol cues in college drinkers. *Alcoholism: Clinical Experimental Research* **37**, E161-E171 (2013).
12. Schweinsburg, A.D., *et al.* An FMRI study of response inhibition in youths with a family history of alcoholism. *ANNALS-NEW YORK ACADEMY OF SCIENCES* **1021**, 391-394 (2004).
13. Bell, R.L., Rodd, Z.A., Lumeng, L., Murphy, J.M. & McBride, W.J. The alcohol-preferring P rat and animal models of excessive alcohol drinking. *Addiction Biology* **11**, 270-288 (2006).
14. Timme, N.M., *et al.* Alcohol preferring P rats exhibit aversion resistant drinking of alcohol adulterated with quinine. *Alcohol* **83**, 47-56 (2020).
15. Ashby, D.M., *et al.* LTD is involved in the formation and maintenance of rat hippocampal CA1 place-cell fields. *Nature Communications* **12**, 100 (2021).
16. Cuzon Carlson, V.C., Gremel, C.M. & Lovinger, D.M. Gestational alcohol exposure disrupts cognitive function and striatal circuits in adult offspring. *Nature Communications* **11**, 2555 (2020).
17. Karadimas, S.K., *et al.* Sensory cortical control of movement. *Nature Neuroscience* **23**, 75-86 (2020).
18. Ottenheimer, D.J., *et al.* A quantitative reward prediction error signal in the ventral pallidum. *Nature Neuroscience* **23**, 1267-1276 (2020).
19. Bice, P.J. & Kiefer, S.W. Taste reactivity in alcohol preferring and nonpreferring rats. *Alcoholism, Clinical and Experimental Research* **14**, 721-727 (1990).

REVIEWER COMMENTS

Reviewer #1 (Remarks to the Author):

I have read through the response to reviewers and the new manuscript. Many of my concerns have been addressed either with new analyses or descriptions/caveats in the manuscript.

I do like the approach and find the work a nice demonstration of population representation and the decoding from PCA but limited from single unit now presented enhances and supports the findings. I think that data is solid.

In relation to point 1 and others tied under the general concern that the authors are looking at a strain difference in firing patterns not driven by compulsive alcohol seeking. The authors clarify that behavior drives the phenotype classification and not genotype. Analyses aimed at examining the switched subjects in behavioral classification (against their genotype group) and whether they “match” the firing patterns (or some measure) of others in the behavioral group would add some evidence against this being merely a strain difference- where different strains differentially change firing patterns (i.e. employ different strategies) to execute a decision/behavior. The two-directional argument of behavioral phenotype on one hand and on the other that there are valid genetic reasons why they would differ leaves this reviewer confused. I understand genetic predisposition- but do worry that these findings could be driven by strain differences independent of anything to do with seeking alcohol per se (no sucrose control). Now whether such a difference would support altered alcohol seeking or consumption is a potential hypothesis and more discussion on this point is warranted.

No additional manipulation study was done- i.e. inhibiting in Wistars to produce compulsive drinking. I am mixed on this. Of course, it would be nice to see if it worked- however if it didn't work I don't think it necessarily rules out the role of these representations. As the authors note in the rebuttal, it could be that broad inhibition would alter many different representations, so tying one specifically to compulsive control would be a stretch. It could be that in P rats these representations, or the lack of engagement is important to the development of the phenotype through interactions with other behaviors, representations, circuits, etc.

Reviewer #2 (Remarks to the Author):

I would like to thank the authors for providing some additional explanations regarding my previous concerns. However, I was hoping that the revised version of this manuscript included the experiment I proposed to strengthen the conclusions. That is, chemogenetic-mediated inhibition of PFC in Wistar rats to produce compulsive drinking.

In the rebuttal, the authors said...

“The ability to restore a pathological behavior is often more valuable from a treatment perspective than generating a pathological state – thus our focus on “treating” the P rats.”

Agree, but from a scientific rigor perspective, inhibition of PFC in Wistars (in which the

increased neural representations believe to prevent compulsive drinking are observed) should be the better test for this hypothesis. Only by inhibiting or preventing those representations from occurring in the Wistar rats will truly test their necessity. The existing findings with excitatory DREADDs in the P-rats are consistent with the hypothesis, but this experiment did not test the necessity for those neural representations.

Reviewer #3 (Remarks to the Author):

The authors have sufficiently addressed my previous comments. I have no further comments.

Response to Reviewers

We would like to thank the reviewers and the editor for taking the time to review our resubmission and provide additional helpful comments so quickly. We greatly appreciate your efforts! We believe we have addressed these concerns in the submission and we list our responses below in red text. Thank you again for all your hard work!

Reviewer #1 (Remarks to the Author):

I have read through the response to reviewers and the new manuscript. Many of my concerns have been addressed either with new analyses or descriptions/caveats in the manuscript.

I do like the approach and find the work a nice demonstration of population representation and the decoding from PCA but limited from single unit now presented enhances and supports the findings. I think that data is solid.

Thank you!

In relation to point 1 and others tied under the general concern that the authors are looking at a strain difference in firing patterns not driven by compulsive alcohol seeking. The authors clarify that behavior drives the phenotype classification and not genotype. Analyses aimed at examining the switched subjects in behavioral classification (against their genotype group) and whether they “match” the firing patterns (or some measure) of others in the behavioral group would add some evidence against this being merely a strain difference- where different strains differentially change firing patterns (i.e. employ different strategies) to execute a decision/behavior.

We appreciate your suggestion to more closely examine the neural behavior of the switched compulsive test data. It would be problematic to examine these recordings alone in the PCA due to several unresolved questions associated with splitting out individual recordings, analyzing them with PCA, and relating them in a meaningful way to the PCA that was conducted with all the recordings. However, we did examine the individual neuron firing rate responses to drinking and found patterns that largely agree with the results observed in the rest of the analysis: increased firing in non-compulsive subjects and no changes in compulsive subjects during quinine adulteration. We believe this result increases confidence in the method of switching these recordings based on phenotypic behavior and the underlying changes in neural activity that underlie compulsive drinking. We have added subfigures to Supplemental Figure 14 to detail these results.

The two-directional argument of behavioral phenotype on one hand and on the other that there are valid genetic reasons why they would differ leaves this reviewer confused. I understand genetic predisposition- but do worry that these findings could be driven by strain differences independent of anything to do with

seeking alcohol per se (no sucrose control). Now whether such a difference would support altered alcohol seeking or consumption is a potential hypothesis and more discussion on this point is warranted.

We were unsure of exactly what you meant in this section. We requested additional information through the editor. This is the additional information you provided:

First- some evidence that the neural representations like that of the P rats are found in specifically the Wistars that they included in the compulsive group (n = 2). I know this is a low n; however, by taking a behavioral phenotype approach (the compulsive seeking) and making a claim about neural representation differences across strains (save the 2 Wistars included) raises the additional explanation of the representation or lack thereof being attributable to strain differences and not driven by compulsive alcohol seeking. At this point, I am not clear as to whether there is any data showing this isn't a strain effect; that these neural representations (or lack thereof) if found in other rats would be related to compulsive seeking. There is not a sucrose group showing whether similar patterns in P rats are found independent of ethanol as a reinforcer, thus the possibility it appears is still open that this is a strain difference.

Thank you for clarifying your point! First, regarding your request for additional analyses of the switched animals, please see our response above. Second, regarding the possibility that the differences we observed in neural computations are strain differences unrelated to compulsive drinking, we agree that this is an important point. We better clarified this point by clearly articulating that this is a possibility before discussing our rationale for why we believe it is unlikely (see lines 595-596 in the track changes version of the manuscript).

No additional manipulation study was done- i.e. inhibiting in Wistars to produce compulsive drinking. I am mixed on this. Of course, it would be nice to see if it worked- however if it didn't work I don't think it necessarily rules out the role of these representations. As the authors note in the rebuttal, it could be that broad inhibition would alter many different representations, so tying one specifically to compulsive control would be a stretch. It could be that in P rats these representations, or the lack of engagement is important to the development of the phenotype through interactions with other behaviors, representations, circuits, etc.

We appreciate you mentioning this point and taking the time to consider our comments on this issue in our previous response to reviewers document. While we agree that it is intuitive to pursue the dmPFC DREADD inhibition study in Wistars to see if it produces the opposite effect (i.e., increased compulsive drinking), after thorough consideration, we believe it would not provide helpful data. We agree with you that if increased inhibition in Wistars did not increase compulsive drinking, it would not impact conclusions about the role of the observed representations in the neurocomputations underlying compulsive drinking. Broad, persistent increases in inhibition caused by activated DREADDs are likely to produce numerous changes to neural function and representations, possibly in unexpected ways. Thus, any resulting changes in behavior, including with regards to compulsive drinking, will be difficult to interpret and could produce misleading results in terms of neural representations, which are the primary focus of the paper. Therefore, we feel that such a study is not scientifically justified. That said,

we believe that there are experiments that could be conducted to examine the functional effects of altered neural excitation and inhibition along with changes in neural representations. For instance, simultaneous temporally precise manipulation of neural firing (e.g., via optogenetic techniques) and neural recordings would provide valuable information about the neurocomputational properties of dmPFC with regards to compulsive drinking. We are currently working to develop these experiments in our lab, but we feel they represent a significant increase in experimental complexity that renders such studies beyond the scope of this work. This already lengthy manuscript lays the foundation for later simultaneous manipulation and recording experiments. To help clarify this point to the readers, we have better described these experiments (see lines 587-590 in the track changes version of the manuscript).

Reviewer #2 (Remarks to the Author):

I would like to thank the authors for providing some additional explanations regarding my previous concerns. However, I was hoping that the revised version of this manuscript included the experiment I proposed to strengthen the conclusions. That is, chemogenetic-mediated inhibition of PFC in Wistar rats to produce compulsive drinking.

In the rebuttal, the authors said...

“The ability to restore a pathological behavior is often more valuable from a treatment perspective than generating a pathological state – thus our focus on “treating” the P rats.”

Agree, but from a scientific rigor perspective, inhibition of PFC in Wistars (in which the increased neural representations believe to prevent compulsive drinking are observed) should be the better test for this hypothesis. Only by inhibiting or preventing those representations from occurring in the Wistar rats will truly test their necessity. The existing findings with excitatory DREADDs in the P-rats are consistent with the hypothesis, but this experiment did not test the necessity for those neural representations.

Thank you for taking the time to consider our resubmission. We respectfully disagree with your view of the DREADD inhibition experiment. *Using DREADDs in Wistars to increase inhibition will produce broad, persistent changes in neural firing that are likely to alter representations in unknown ways.* In other words, it is not clear that inhibitory DREADDs would selectively silence the representations we observed using electrophysiology. Thus, any resulting changes in behavior, including changes in compulsive drinking, will be difficult to interpret and could be misleading in terms of neural representations, which are the primary focus of the paper. Therefore, we feel that such a study is not scientifically justified.

That said, we believe that there experiments that could be conducted to examine the functional effects of altered neural excitation and inhibition along with changes in neural representations. For instance, simultaneous temporally precise manipulation of neural firing (e.g., via optogenetic techniques) and neural recordings could provide valuable information about the neurocomputational properties of dmPFC with regards to compulsive drinking. We are currently working to develop these experiments in our lab, but we feel they represent a significant increase in experimental complexity that renders such

studies beyond the scope of this work. This already lengthy manuscript lays the foundation for later simultaneous manipulation and recording experiments. To help clarify this point to the readers, we have better described these experiments (see lines 587-590 in the track changes version of the manuscript).

REVIEWERS' COMMENTS

Reviewer #1 (Remarks to the Author):

The authors have addressed my concerns